# Permutation Equivariant Neural Networks for Antisymmetric Tensors

## Abstract

Antisymmetric tensors, which change sign under index swaps, appear naturally in physics, yet learning from them remains largely unexplored. We provide a complete characterisation of all linear permutation equivariant functions between antisymmetric power spaces of $\mathbb{R}^n$. To make this characterisation practical, we introduce a memory-efficient implementation that eliminates the need to create and store large weight matrices. We demonstrate that our approach is efficient in learning functions that depend on the antisymmetric structure of the input and outperforms models that do not incorporate this structure as an inductive bias.

## 1 Introduction

Equivariance provides a principled framework for designing neural networks that respect symmetries in data. By constraining functions to commute with group actions, equivariant models encode structural priors, which can improve generalisation and sample efficiency. This idea was first introduced in the context of group equivariant convolutional neural networks (Cohen & Welling, 2016; 2017) and has since been applied to more general group actions on a variety of structured data domains (Kondor & Trivedi, 2018; Kondor et al., 2018; Thomas et al., 2018; Weiler et al., 2018; Satorras et al., 2021). A central challenge in equivariant machine learning is the characterisation of linear equivariant maps between group representation spaces, since they form the fundamental building blocks for developing new equivariant architectures (Ravanbakhsh et al., 2017; Zaheer et al., 2017; Maron et al., 2019a; Finzi et al., 2021; Pearce-Crump, 2023; Godfrey et al., 2023; Sverdlov et al., 2025).

Despite these advances, equivariant modelling of certain classes of structured data remains comparatively less explored. In particular, antisymmetric tensors, whose components satisfy

$$T_{i_1,i_2,\ldots,i_k} = \text{sgn}(\pi)T_{i_{\pi(1)},i_{\pi(2)},\ldots,i_{\pi(k)}} \qquad \text{for all } \pi \in S_k \tag{1}$$

and vanish whenever any two indices coincide, present a unique challenge that existing equivariant methods do not address. These tensors arise in areas of physics and mathematics, including differential geometry (Lee, 2009; Tu, 2017), electromagnetism (Misner et al., 1973), general relativity (Wald, 1984), and quantum mechanics (Bloch & Messiah, 1962). Many key operations on these tensors, such as computing exterior derivatives in differential geometry or applying operators to fermionic states in quantum mechanics, are linear and must be permutation equivariant in order to preserve antisymmetry. However, existing permutation equivariant architectures (Zaheer et al., 2017; Maron et al., 2019a; Godfrey et al., 2023; Pearce-Crump, 2024) are designed for general tensors and do not guarantee the preservation of the antisymmetric structure, limiting their effectiveness on such data.

In this work, we address this gap by making the following contributions:

- **Exact characterisation**: We determine a complete characterisation of all permutation equivariant linear maps between any two antisymmetric powers of $\mathbb{R}^n$, covering both scalar- and vector-valued functions.
- **Efficient implementation**: We design a memory-efficient approach, which we term IO patterns, that avoids the explicit construction of large weight matrices while enabling permutation equivariant mappings between antisymmetric tensors of arbitrary order.
- **Empirical evaluation**: We evaluate our framework on two tasks where the target function depends on the antisymmetric structure of the input, showing that it outperforms standard MLPs and standard permutation equivariant neural networks on these tasks.

## 2 RELATED WORK

**Equivariant Machine Learning on Antisymmetric Tensors**   Incorporating equivariance into learning from antisymmetric tensors is a relatively new area of research. To the best of our knowledge, only a few recent publications have addressed this topic, but they focus only on equivariance with respect to Euclidean groups. Heilman et al. (2024) created $SO(3)$-equivariant models using spherical harmonics to predict components of antisymmetric tensors from crystal structures. Hodapp & Shapeev (2024) created equivariant tensor networks to construct interatomic interaction models that are based on $SO(3)$-invariant tensor networks. By contrast, our work explores permutation equivariance on antisymmetric tensors.

**Equivariant Machine Learning on Symmetric Tensors**   Several works in equivariant machine learning have studied structured data in the symmetric tensor setting (Gao et al., 2022; Lou & Ganose, 2024; Wen et al., 2024; Heilman et al., 2024; Garanger et al., 2024). Pearce-Crump (2025) also characterised permutation equivariant linear maps for symmetric tensors. Their framework, however, cannot handle antisymmetric tensors because the signs of permutations that must be considered in equation (6) create a fundamentally different combinatorial structure. A key contribution of our work is the identification of valid basis elements for permutation equivariant linear maps between antisymmetric tensors, providing a complete characterisation in this setting. This also enables a compact, memory-efficient implementation that is not possible with previous approaches.

**Permutation Equivariant Neural Networks**   These neural networks have been designed to learn from different types of data, focusing primarily on set-structured data (Ravanbakhsh et al., 2017; Qi et al., 2017; Zaheer et al., 2017; Hartford et al., 2018; Maron et al., 2020; Sverdlov et al., 2025) and graph- and hypergraph-structured data (Maron et al., 2019a; Thiede et al., 2020; Finzi et al., 2021; Morris et al., 2022; Godfrey et al., 2023; Puny et al., 2023; Pearce-Crump, 2024; Pearce-Crump & Knottenbelt, 2024b). In particular, Pan & Kondor (2022) and Pearce-Crump & Knottenbelt (2024a) have investigated how to organise the computations that are involved in these networks to improve their performance. Additionally, a separate line of research has explored their expressive power (Keriven & Peyré, 2019; Maron et al., 2019b; Segol & Lipman, 2020; Yarotsky, 2022).

## 3 ANTISYMMETRIC POWERS OF $\mathbb{R}^n$

We present the fundamental definitions that are needed to be able to study linear permutation equivariant functions on antisymmetric tensors. We first introduce an action of $S_n$ on a set of $k$-length indices $\Lambda[n]^k$. These indices will then index a basis of a real vector space, $\Lambda^k(\mathbb{R}^n)$, known as the $k^{\text{th}}$ antisymmetric power of $\mathbb{R}^n$. We let $[n] \coloneqq \{1, \ldots, n\}$ throughout.

Define the set $\Lambda[n]^k$ to consist of all $k$-length tuples of the form

$$I = (i_1, i_2, \ldots, i_k) \text{ such that } 1 \leq i_1 < i_2 < \ldots < i_k \leq n. \tag{2}$$

We see that, for $k \leq n$, the cardinality of $\Lambda[n]^k$ is $\binom{n}{k}$, otherwise it is 0.

We obtain an action of the symmetric group $S_n$ on $\Lambda[n]^k$ by applying the permutation to each element in the tuple and then reordering the result (applying a permutation in $S_k$) such that the lexicographical ordering is maintained.

*Example* 3.1. Consider the element $(1, 3, 5)$ of $\Lambda[5]^3$, and suppose that we apply $(145)(23)$ in $S_5$ to it. We obtain $(4, 2, 1)$ which we reorder to $(1, 2, 4)$ to give another element of $\Lambda[5]^3$.

Consequently, we define the **$k^{\text{th}}$ antisymmetric power of** $\mathbb{R}^n$, $\Lambda^k(\mathbb{R}^n)$, to be the vector space that has a **standard basis** $\{e_I\}$ indexed by the elements $I$ of $\Lambda[n]^k$. Typically one writes $e_I \in \Lambda^k(\mathbb{R}^n)$ as $e_{i_1} \wedge e_{i_2} \wedge \cdots \wedge e_{i_k}$, where $\wedge$ denotes the **wedge product**.

We can also raise the action of $S_n$ on the set $\Lambda[n]^k$ to a representation of $S_n$ on $\Lambda^k(\mathbb{R}^n)$ in the usual way — however, in applying the permutation (in $S_k$) that maintains the lexicographical ordering of the indices, we need to scalar multiply the result by the sign of this permutation. Denoting the representation itself by $\rho_k^\Lambda$, we therefore see that

$$\rho_k^\Lambda(\sigma) : e_I \mapsto \text{sgn}(\pi_{\sigma(I),k}) e_{\sigma(I)\pi_{\sigma(I),k}} \tag{3}$$

where $\pi_{\sigma(I),k}$ is the permutation in $S_k$ that maintains the lexicographical ordering of $\sigma(I)$.

*Example* 3.2. Following on from Example 3.1, we see that

$$\rho_3^\Lambda((145)(23)) : e_{(1,3,5)} \mapsto -e_{(1,2,4)} \tag{4}$$

since the permutation in $S_3$ that reorders $(4, 2, 1)$ to $(1, 2, 4)$, namely $(13)$, has sign $-1$.

We wish to characterise in full the linear permutation equivariant functions between any two antisymmetric power spaces of $\mathbb{R}^n$. They are defined as follows.

**Definition 3.3.** A linear map $\phi : \Lambda^k(\mathbb{R}^n) \to \Lambda^l(\mathbb{R}^n)$ is said to be **permutation equivariant** if, for all $\sigma \in S_n$ and $v \in \Lambda^k(\mathbb{R}^n)$,

$$\phi(\rho_k^\Lambda(\sigma)[v]) = \rho_l^\Lambda(\sigma)[\phi(v)] \tag{5}$$

We denote the vector space of all such maps by $\mathrm{Hom}_{S_n}(\Lambda^k(\mathbb{R}^n), \Lambda^l(\mathbb{R}^n))$.

Since our goal is to calculate all of the permutation equivariant weight matrices that can appear between any two antisymmetric power spaces of $\mathbb{R}^n$, it is enough to construct a basis of matrices for $\mathrm{Hom}_{S_n}(\Lambda^k(\mathbb{R}^n), \Lambda^l(\mathbb{R}^n))$, by viewing it as a subspace of $\mathrm{Hom}(\Lambda^k(\mathbb{R}^n), \Lambda^l(\mathbb{R}^n))$ and choosing the standard basis for each antisymmetric power space, since any weight matrix will be a weighted linear combination of these basis matrices. This characterisation includes the following functions:

**Proposition 3.4.** *Linear permutation equivariant scalar-valued and vector-valued functions on antisymmetric tensors in $(\mathbb{R}^n)^{\otimes k}$ are elements of $\mathrm{Hom}_{S_n}(\Lambda^k(\mathbb{R}^n), \Lambda^0(\mathbb{R}^n))$ and $\mathrm{Hom}_{S_n}(\Lambda^k(\mathbb{R}^n), \Lambda^1(\mathbb{R}^n))$, respectively.*

## 4 BASIS OF $\mathrm{Hom}_{S_n}(\Lambda^k(\mathbb{R}^n), \Lambda^l(\mathbb{R}^n))$

We obtain the basis of matrices for $\mathrm{Hom}_{S_n}(\Lambda^k(\mathbb{R}^n), \Lambda^l(\mathbb{R}^n))$ in three parts: firstly, we obtain a condition that a basis matrix needs to satisfy; then we raise this condition to the level of orbits of $S_n$ acting on the Cartesian product set $\Lambda[n]^l \times \Lambda[n]^k$, equating each orbit with a certain type of diagram; and finally we use these diagrams to obtain the basis matrices themselves.

**Proposition 4.1.** *The basis elements $f$ of $\mathrm{Hom}_{S_n}(\Lambda^k(\mathbb{R}^n), \Lambda^l(\mathbb{R}^n))$ must satisfy*

$$\boxed{f_{I,J} = \mathrm{sgn}(\pi_{\sigma(I),l})\, \mathrm{sgn}(\pi_{\sigma(J),k}) f_{\sigma(I)\pi_{\sigma(I),l}, \sigma(J)\pi_{\sigma(J),k}}} \tag{6}$$

*for all $\sigma \in S_n$, $I \in \Lambda[n]^l$ and $J \in \Lambda[n]^k$, where $\pi_{\sigma(I),l}$ and $\pi_{\sigma(J),k}$ are permutations in $S_l$ and $S_k$ that restore the lexicographical ordering of $\sigma(I)$ and $\sigma(J)$, respectively.*

Equation (6) suggests that we can find the basis elements of $\mathrm{Hom}_{S_n}(\Lambda^k(\mathbb{R}^n), \Lambda^l(\mathbb{R}^n))$ by studying the action of $S_n$ on the Cartesian product set $\Lambda[n]^l \times \Lambda[n]^k$. Indeed, if we write the elements of $\Lambda[n]^l \times \Lambda[n]^k$ as $\binom{I}{J}$, then we can define an action of $S_n$ on $\Lambda[n]^l \times \Lambda[n]^k$ by

$$\sigma\binom{I}{J} := \binom{\sigma(I)\pi_{\sigma(I),l}}{\sigma(J)\pi_{\sigma(J),k}} \tag{7}$$

In fact, instead of studying the orbits under this action, we show that each orbit corresponds to a diagram which we will then use to obtain the basis elements of $\mathrm{Hom}_{S_n}(\Lambda^k(\mathbb{R}^n), \Lambda^l(\mathbb{R}^n))$.

We start by constructing an equivalence class of diagrams for each orbit coming from the action of $S_n$ on $\Lambda[n]^l \times \Lambda[n]^k$. Let $\binom{I}{J}$ be a class representative of some orbit. We form a diagram for $\binom{I}{J}$ in the following way:

> We place the values of $I$ in one row and the values of $J$ in another row below it. Then, for each value $x \in [n]$, if $x$ does not appear in either $I$ or $J$, we move onto the next value in $[n]$. Otherwise, we insert a central red node in between the two rows. If $x$ appears in $I$, then we draw a line from its position in the top row down to the red node. If $x$ appears in $J$, then we draw a line from its position in the bottom row up to the red node.

*Example* 4.2. Suppose that $l = 3$, $k = 3$, and $n = 5$. Consider the $S_5$ orbit of $\Lambda[5]^3 \times \Lambda[5]^3$ that contains the element $\binom{125}{123}$. Then, by the construction given in the green-coloured box, we obtain

$$\tag{8}$$

Removing the labels gives the resulting diagram for the element $\binom{125}{123}$. In general, for such a diagram, we call the black lines **wires** and the individual central red nodes together with the wires that are attached to them **spiders**.

Suppose that we change the class representative of the orbit: if we choose $\binom{\sigma(I)\pi_{\sigma(I),l}}{\sigma(J)\pi_{\sigma(J),k}}$ instead and form a diagram using the same procedure given in the green-coloured box above, then the spiders that appear in the diagram for this class representative (viewed without their labels) will be the same as the one for $\binom{I}{J}$, just potentially in a different order. Hence, by our construction, for each orbit coming from the action of $S_n$ on $\Lambda[n]^l \times \Lambda[n]^k$, we obtain an equivalence class of diagrams that corresponds to it, as desired.

*Example* 4.3. Continuing on from Example 4.2, we see that $\binom{235}{135}$ is in the same orbit as $\binom{125}{123}$, since it is obtained by first applying the permutation $(13)(25)$ to $\binom{125}{123}$ and then restoring the lexicographical ordering by applying separate permutations in $S_3$ to each of the top and bottom rows of the diagram. Hence, applying the procedure given in the green-coloured box above to $\binom{235}{135}$, we obtain the diagram

$$\tag{9}$$

Note that, after removing the labels, this gives the same diagram as (8), except the spiders are in a different order.

However, we can say more, once we have introduced the following concept:

**Definition 4.4.** A $(k, l)$**–bipartition** $\pi$ is the set

$$\{[x_1, y_1], [x_2, y_2], \ldots, [x_t, y_t]\} \tag{10}$$

of some $t$ pairs $[x_i, y_i]$ such that $x_i, y_i \geq 0$ for all $i$, not both zero, $\sum_{i=1}^{t} x_i = k$ and $\sum_{i=1}^{t} y_i = l$. We call the individual pairs **blocks**. Bipartitions were first introduced and studied by MacMahon (1896).

We define a particular type of bipartition as follows. To the best of our knowledge, they have not appeared in the literature before, hence we have chosen the most appropriate name that we could think of for them.

**Definition 4.5.** We call a $(k, l)$–bipartition $\pi$ where each block is restricted to being only either $[1, 1]$, $[1, 0]$, or $[0, 1]$ a $(k, l)$**–antispherical bipartition**.

Hence we see that, for each orbit coming from the action of $S_n$ on $\Lambda[n]^l \times \Lambda[n]^k$, every diagram in the equivalence class (of diagrams) that corresponds to the orbit represents the *same* $(k, l)$–antispherical bipartition since the three possible types of blocks in a $(k, l)$–antispherical bipartition correspond to the following spiders:

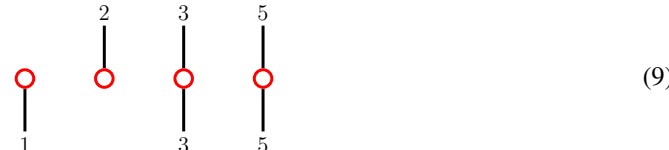

$$\tag{11}$$

We call the diagrams $(k, l)$**–antispherical bipartition diagrams**. Consequently, we can choose a particular antispherical bipartition diagram to represent the entire equivalence class, and hence it is the diagram that corresponds to the orbit:

**Definition 4.6.** The **normal form** of a $(k, l)$–antispherical bipartition is the $(k, l)$–antispherical bipartition diagram where, from left to right, all of the $[1, 1]$ parts are represented as spiders, then all of the $[1, 0]$ parts, and finally all of the $[0, 1]$ parts.

*Example* 4.7. Continuing on from Examples 4.2 and 4.3, we see that the $(3, 3)$–antispherical bipartition that corresponds to the orbit containing both $\binom{125}{123}$ and $\binom{235}{135}$ is

$$\{[1, 1], [1, 1], [1, 0], [0, 1]\} \tag{12}$$

The corresponding $(3, 3)$–antispherical bipartition in normal form is given in (8) (without the labels).

*Remark* 4.8. We note that the $(k, l)$–antispherical bipartition that corresponds to an orbit can have at most $n$ blocks, by the construction given in the green-coloured box.

Having shown that each orbit coming from the action of $S_n$ on $\Lambda[n]^l \times \Lambda[n]^k$ corresponds to (the normal form of) a $(k, l)$–antispherical bipartition having at most $n$ blocks we now show that all $(k, l)$–antispherical bipartitions having at most $n$ blocks must appear in this process.

Representing each $(k, l)$–antispherical bipartition having some $t \in [n]$ blocks in its normal form, we obtain its orbit by following the six steps below:

1. Form all possible $t$-length tuples with elements in $[n]$, not allowing for repetitions amongst the elements.

2. Then, for each $t$-length tuple, label the central red nodes of the normal form from left to right with the elements of the tuple.

3. If necessary, reorder the spiders so that they are in increasing order from left to right.

4. Then, for each spider, propagate the block label to the end of each wire.

5. This produces the element $\binom{I}{J}$, where the top row's labels are $I$ and the bottom row's labels are $J$.

6. Doing this for each $t$-length tuple gives the entire $S_n$ orbit.

Hence, overall, we have shown the following result:

**Proposition 4.9.** *The orbits that come from the action of $S_n$ on $\Lambda[n]^l \times \Lambda[n]^k$ correspond bijectively with (the normal form of) the $(k, l)$–antispherical bipartitions having at most $n$ blocks.*

*Remark* 4.10. In the Technical Appendix C, we provide a novel procedure for how to generate all $(k, l)$–antispherical bipartitions having at most $n$ blocks, and hence determine their total number.

We now describe how to obtain the basis matrices of $\operatorname{Hom}_{S_n}(\Lambda^k(\mathbb{R}^n), \Lambda^l(\mathbb{R}^n))$ from $(k, l)$–antispherical bipartitions having at most $n$ blocks.

Looking back at equation (6) and applying Proposition 4.9, we see that, in order to find the basis elements of $\operatorname{Hom}_{S_n}(\Lambda^k(\mathbb{R}^n), \Lambda^l(\mathbb{R}^n))$, we not only need to consider all $(k, l)$–antispherical bipartitions having at most $n$ blocks, but we also need to keep track of the signs of the individual permutations $\pi_{\sigma(I),l}$ and $\pi_{\sigma(J),k}$ that restore the lexicographical ordering of $\sigma(I)$ and $\sigma(J)$ respectively. This means that for each $(k, l)$–antispherical bipartition having at most $n$ blocks, we need to choose some $f_{I,J}$ to have a positive sign and calculate the signs of each $f_{\sigma(I)\pi_{\sigma(I),l},\sigma(J)\pi_{\sigma(J),k}}$ with respect to this initial choice.

This is why we introduced the normal form of a $(k, l)$–antispherical bipartition, namely to achieve consistency in how we calculate these signs as we act with the symmetric group $S_n$. Indeed, suppose that the normal form of a $(k, l)$–antispherical bipartition has some $t \in [n]$ blocks. Then we choose the diagram that comes from labelling the $t$ blocks of the normal form from left to right with the elements of the tuple $(1, 2, \ldots, t)$ to have a positive sign (and hence we can also assign the same positive sign to its associated $\binom{I}{J}$ element that comes from propagating the block labels to the ends of the wires). After making this choice, we then label the normal form with each possible $t$-length tuple having distinct elements in $[n]$. For each such labelled diagram, we need to associate a sign with the diagram that is obtained by reordering the labelled spiders so that they are in increasing order from left to right — let us call this the reordered diagram. This is because labelling the central nodes of the normal form with the elements of some $t$-length tuple is equivalent to acting on the $I, J$ tuples in the chosen $\binom{I}{J}$ element with some permutation $\sigma$, and reordering the spiders restores the lexicographical

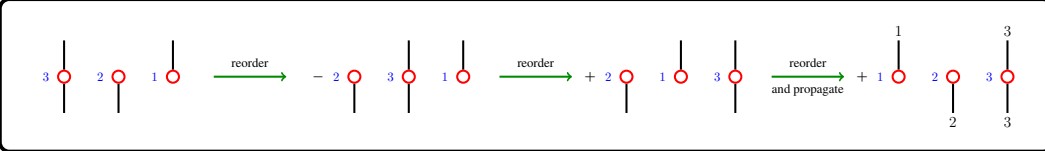

Figure 1: We show how to calculate the sign of the reordered diagram that comes from the normal form of the $(2, 2)$–antispherical bipartition $\{[1, 1], [1, 0], [0, 1]\}$ whose spiders are labelled with the 3-length tuple $(3, 2, 1)$. We calculate the signs either by tracking the swaps that we make in the top row and the bottom row or by following the rules that are given in Figure 2.

ordering in both $\sigma(I)$ and $\sigma(J)$. That is, each reordered diagram corresponds to the $\binom{\sigma(I)\pi_{\sigma(I),l}}{\sigma(J)\pi_{\sigma(J),k}}$ element in the associated orbit (by propagating the block labels to the ends of the wires). Hence, associating a sign to each such reordered diagram is the same as associating a sign to its associated $\binom{\sigma(I)\pi_{\sigma(I),l}}{\sigma(J)\pi_{\sigma(J),k}}$ element.

For each diagram that is obtained by labelling the normal form with some $t$-length tuple, we associate a sign to its reordered diagram as follows: we begin with the normal form that is labelled with the $t$-length tuple, which has positive sign, and, as we swap adjacent pairs of spiders, we track the swaps that we make in the top row and the bottom row. If the result consists of only one swap in either of the top or bottom rows, we change the sign, otherwise we leave the sign as it is. The sign that appears next to the diagram once it is reordered is the one that is associated with the reordered diagram.

*Example* 4.11. We give an example of this procedure in Figure 1: we show how to calculate the sign of the reordered diagram that comes from the normal form of the $(2, 2)$–antispherical bipartition $\{[1, 1], [1, 0], [0, 1]\}$ whose spiders are labelled with the 3-length tuple $(3, 2, 1)$.

We can go further: since we are tracking sign changes as we swap adjacent pairs of spiders, we can view the sign changes as a set of rules on pairs of spiders as we swap their order. Reducing these swaps to a set of rules means that we do not have to track swaps on individual rows. We have provided these rules in Figure 2. Their proof is immediate from equation (6) together with the fact that the sign of a transposition is $-1$.

We have resisted saying throughout that, for a given $(k, l)$–antispherical bipartition having at most $n$ blocks, once we obtain a sign for each reordered diagram (and hence for each $\binom{\sigma(I)\pi_{\sigma(I),l}}{\sigma(J)\pi_{\sigma(J),k}}$), then we know the sign of the $(\sigma(I)\pi_{\sigma(I),l}, \sigma(J)\pi_{\sigma(J),k})$–entry of its associated basis matrix. This is because certain classes of $(k, l)$–antispherical bipartitions having at most $n$ blocks do not generate a basis matrix: in fact, they generate the zero matrix.

**Lemma 4.12.** *Any $(k, l)$–antispherical bipartition having at most $n$ blocks that has more than one $[1, 0]$ block or more than one $[0, 1]$ block corresponds to the zero matrix in $\mathrm{Hom}(\Lambda^k(\mathbb{R}^n), \Lambda^l(\mathbb{R}^n))$.*

However, with the following theorem, for the remaining $(k, l)$–antispherical bipartitions having at most $n$ blocks, once we have obtained a sign for each reordered diagram, then we *do* know what the sign for the $(\sigma(I)\pi_{\sigma(I),l}, \sigma(J)\pi_{\sigma(J),k})$–entry of its associated basis matrix will be. (The proof shows that, for these $(k, l)$–antispherical bipartitions, all reordered diagrams that generate the same $\binom{\sigma(I)\pi_{\sigma(I),l}}{\sigma(J)\pi_{\sigma(J),k}}$ element come with the same sign). Hence, if the sign is positive, then we put $+1$ in the $(\sigma(I)\pi_{\sigma(I),l}, \sigma(J)\pi_{\sigma(J),k})$–entry of its associated basis matrix, otherwise we put $-1$. We fill all other entries with 0.

**Theorem 4.13.** *The basis matrices $X_\pi$ of $\mathrm{Hom}_{S_n}(\Lambda^k(\mathbb{R}^n), \Lambda^l(\mathbb{R}^n))$ are in bijective correspondence with the set of $(k, l)$–antispherical bipartitions $\pi$ having at most $n$ blocks such that each antispherical bipartition has at most one $[1, 0]$ block and at most one $[0, 1]$ block.*

**Corollary 4.14.** *The $S_n$-equivariant weight matrix from $\Lambda^k(\mathbb{R}^n)$ to $\Lambda^l(\mathbb{R}^n)$ is $\sum_\pi w_\pi X_\pi$ for weights $w_\pi$ and basis matrices $X_\pi$, where the sum is over all $(k, l)$–antispherical bipartitions $\pi$ having at most $n$ blocks with at most one $[1, 0]$ block and at most one $[0, 1]$ block.*

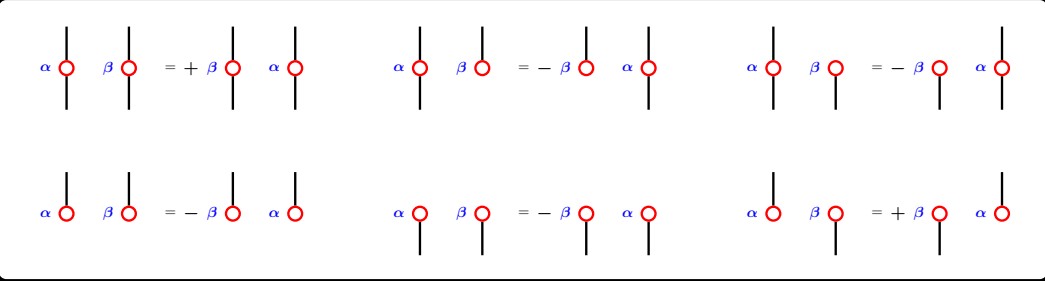

Figure 2: This set of six rules describes the resulting sign changes that occur when we swap the order of a pair of labelled spiders. Swaps that consist of only one swap in either of the top or bottom rows induce an overall change of sign, whereas swaps that consist of either zero or one swap in both of the top and bottom rows retain the existing sign.

*Example* 4.15. Continuing on from Example 4.11, we see that the the $(13, 23)$–entry of the basis matrix $\text{Hom}_{S_n}(\Lambda^2(\mathbb{R}^n), \Lambda^2(\mathbb{R}^n))$, for any $n \geq 3$, corresponding to the $(2, 2)$–bipartition $\pi = \{[1, 1], [1, 0], [0, 1]\}$ is $+1$.

*Example* 4.16. Suppose that we would like to find the $S_3$-equivariant weight matrix from $\Lambda^2(\mathbb{R}^3)$ to $\Lambda^1(\mathbb{R}^3)$. We need to consider the normal form of the $(2, 1)$–antispherical bipartitions having at most $n = 3$ blocks, with at most one $[1, 0]$ block and at most one $[0, 1]$ block. There is only one such diagram:

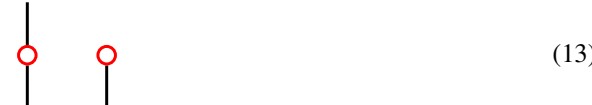

$$\tag{13}$$

This diagram corresponds to a basis matrix in $\text{Hom}_{S_3}(\Lambda^2(\mathbb{R}^3), \Lambda^1(\mathbb{R}^3))$ of size $3 \times 3$. The calculations to obtain this matrix are given in Example D.1. Hence, for a weight $w \in \mathbb{R}$, the $S_3$-equivariant weight matrix from $\Lambda^2(\mathbb{R}^3)$ to $\Lambda^1(\mathbb{R}^3)$ is of the form

$$
\begin{array}{c}
\phantom{1}\begin{array}{ccc} {\color{blue}1,2} & {\color{blue}1,3} & {\color{blue}2,3} \end{array} \\
\begin{array}{c} {\color{blue}1} \\ {\color{blue}2} \\ {\color{blue}3} \end{array}
\left[ \begin{array}{ccc}
w & w & 0 \\
-w & 0 & w \\
0 & -w & -w
\end{array} \right]
\end{array}
\tag{14}
$$

*Remark* 4.17. We have focused solely on understanding the possible $S_n$-equivariant weight matrices from $\Lambda^k(\mathbb{R}^n)$ to $\Lambda^l(\mathbb{R}^n)$ because it is clear that the pointwise non-linearities must be odd in order to preserve the antisymmetric structure.

## 5 MEMORY-EFFICIENT IMPLEMENTATION

Since we would like the weight matrices that we have characterised to operate on antisymmetric tensors in $(\mathbb{R}^n)^{\otimes k}$, we show in the Technical Appendix, Section B, that we can represent any $S_n$-equivariant weight matrix $W$ from $\Lambda^k(\mathbb{R}^n)$ to $\Lambda^l(\mathbb{R}^n)$ as a matrix from $(\mathbb{R}^n)^{\otimes k}$ to $(\mathbb{R}^n)^{\otimes l}$. We also provide a general procedure for obtaining this weight matrix in the Technical Appendix. We call the resulting matrix from $(\mathbb{R}^n)^{\otimes k}$ to $(\mathbb{R}^n)^{\otimes l}$ the **unrolled** $S_n$-equivariant weight matrix, and the process of representing the original matrix in this way **unrolling**.

In reality, it is somewhat painful to obtain an unrolled $S_n$-equivariant weight matrix in this way. More importantly, as we increase the values of $n$, $k$ and $l$, it becomes impractical to store the entire unrolled weight matrix in memory, particularly when we look to implement the transformation of an input tensor $T$ by an unrolled weight matrix $W$.

Instead, we introduce some new notation that describes, for each basis matrix that appears in $W$, the transformation of an input $T \in (\mathbb{R}^n)^{\otimes k}$ into its output $W(T) \in (\mathbb{R}^n)^{\otimes l}$. The key idea behind this construction is that we want to understand, for each fixed index $I = (i_1, i_2, \ldots, i_l)$ of the output $W(T)$, all possible inputs that map to this output without having to form an $n^l \times n^k$ matrix.

**Definition 5.1.** Let $\pi$ be a $(k, l)$–antispherical bipartition. We define an **IO pattern** as follows: for a fixed index $I = (i_1, i_2, \ldots, i_l)$ as output, we calculate all possible ways in which we can label the spiders of the normal form of the $(k, l)$–antispherical bipartition together with the permutations of the spiders for a given labelling that create new inputs which map to this fixed output $I$.

We obtain a pattern for each such combination of the form

$$I \leftarrow J \tag{15}$$

where we sum over all indices in $J$ that do not appear in $I$.

Finally, we combine with addition all of the patterns on the right hand side to obtain the IO pattern itself. We can then use the left hand side of the IO pattern to index the output $X_\pi(T)$ and the right hand side to index the input $T$, which ultimately describes the transformation in full. In this way, we avoid having to create a weight matrix at all, and consequently the only requirements on memory come from needing to store only the input tensor $T$ and the output tensor $W(T)$.

*Example* 5.2. Continuing on from Example 4.16, we consider once again the normal form of the only $(2, 1)$–antispherical bipartition having at most $n = 3$ blocks with at most one $[1, 0]$ block and at most one $[0, 1]$ block. This diagram was given in (13).

Let us fix $i$ as the output index. Then we see that we can obtain this fixed output either from:

$$\text{(16)}$$

By propagating the block labels to the ends of the wires, these diagrams give the patterns

$$i \leftarrow \sum_{j:j\neq i}^{3} i, j \qquad \text{and} \qquad i \leftarrow - \sum_{j:j\neq i}^{3} j, i \tag{17}$$

Combining them we obtain the IO pattern corresponding to this antispherical bipartition:

$$i \leftarrow \sum_{j:j\neq i}^{3} i, j - \sum_{j:j\neq i}^{3} j, i \tag{18}$$

One can check (18) against the basis matrix that is given in Example B.1. Moreover, (18) gives us

$$X_\pi(T)_i = \sum_{j:j\neq i}^{3} T_{i,j} - \sum_{j:j\neq i}^{3} T_{j,i} \tag{19}$$

which describes the transformation on an input antisymmetric tensor $T \in (\mathbb{R}^3)^{\otimes 2}$.

*Remark* 5.3. In fact, there is significant additional power behind the IO pattern method. Namely, with just the one IO pattern, we can generate what would be the corresponding unrolled basis matrix (if we wanted to calculate it) *for any $n$ that is greater than or equal to the number of blocks in the $(k, l)$–antispherical bipartition*, simply by changing the upper limit of summation in each sum that appears in the IO pattern to $n$. This means that the IO pattern encapsulates a whole class of possible transformations (as the value of $n$ changes). See Example 5.2 for an example of this in action.

*Remark* 5.4. In the Technical Appendix, Section F, we have included some more examples of IO patterns for different $(k, l)$–antispherical bipartitions.

## 6 NUMERICAL EXPERIMENTS

We now evaluate our approach on tasks where the target function depends on the antisymmetric structure of the input.

The implementation details relating to the architectures that were chosen and the training details that were used in each task are provided in the Technical Appendix, Section G.

$S_8$-**Equivariant Task:** We evaluate our model on a synthetic $S_8$-equivariant task from antisymmetric tensors in $(\mathbb{R}^8)^{\otimes 2}$ to $\mathbb{R}^8$ that is given by the function $f(T)_i = \sigma\left(\sum_{j:j\neq i} T_{i,j}\right)$ for each $i \in [8]$, where $\sigma$ is the ReLU activation function.

In Figure 3, we show that our model is the only one that is capable of learning the target function, in contrast to both a standard MLP and a standard $S_8$-equivariant model. We attribute this success to the stronger inductive bias that is built into our architecture. We also show in Table 1 that our model appears to generalise well to antisymmetric tensors having different sizes, even though it was only trained on antisymmetric tensors of a particular size.

| $n$ | Test MSE |
|---|---|
| 8 | $0.00003_{\pm 0.000003}$ |
| 24 | $0.0013_{\pm 0.00009}$ |
| 64 | $0.031_{\pm 0.00144}$ |

Table 1: Generalisation.

$S_4$-**Invariant Task:**   We evaluate our model on an $S_4$-invariant task to learn the absolute value of the Pfaffian of $4 \times 4$ antisymmetric matrices. The Pfaffian is important in physics and chemistry as it describes antisymmetric interactions in fermionic systems, such as the electronic structure of molecules. Although this experiment uses synthetic data, it reflects the kind of structured, antisymmetric input that is found in many real-world applications. In Table 2, we show that our model outperforms a standard MLP and a standard $S_4$-invariant model across all training data sizes. In particular, our model achieves the lowest test errors with far fewer training samples and with fewer parameters. We attribute the model's superior performance across all data sizes to its strong inductive bias.

| Model | $3 \cdot 10^1$ | $3 \cdot 10^2$ | $3 \cdot 10^3$ | $3 \cdot 10^4$ |
|---|---|---|---|---|
| **AntiSymmPermEquiv** | $\mathbf{0.3227}_{\pm 0.1146}$ | $\mathbf{0.2735}_{\pm 0.0230}$ | $\mathbf{0.2725}_{\pm 0.0229}$ | $\mathbf{0.2695}_{\pm 0.0222}$ |
| PermEquiv | $0.5307_{\pm 0.0709}$ | $0.2974_{\pm 0.0258}$ | $0.2876_{\pm 0.0229}$ | $0.2756_{\pm 0.0223}$ |
| MLP | $0.7479_{\pm 0.0353}$ | $0.5940_{\pm 0.0899}$ | $0.4665_{\pm 0.0257}$ | $0.3015_{\pm 0.0294}$ |

Table 2: Test MSE for the $S_4$-Invariant Task.

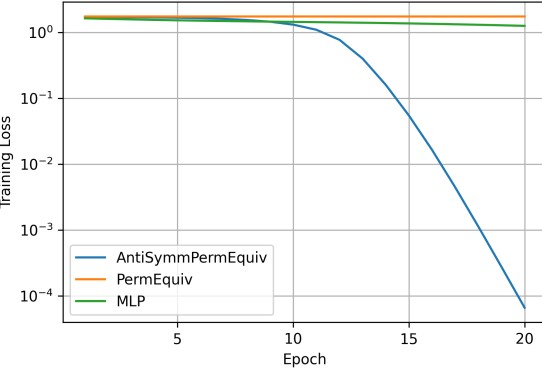

Figure 3: In the $S_8$-equivariant task, the only model that successfully learns the target function is the permutation equivariant one that considers the antisymmetric structure in the input.

## 7  CONCLUSION

In this paper, we obtained a complete characterisation of all linear permutation equivariant functions between antisymmetric powers of $\mathbb{R}^n$. To implement these transformations efficiently, we introduced IO patterns, which encode, for each index in the output tensor, all indices in the input tensor that contribute to the value at this output index. This made it possible for us to perform the transformations without constructing or storing high-dimensional weight matrices. We showed in practical experiments that our network learns better than a standard MLP and a standard permutation equivariant neural network when the target depends on the antisymmetric structure. We also demonstrated the potential for transfer learning to antisymmetric tensors of different sizes.

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

## A  SUPPLEMENTARY PROOFS

*Proof of Proposition 4.1.* Since the vector space $\mathrm{Hom}(\Lambda^k(\mathbb{R}^n), \Lambda^l(\mathbb{R}^n))$ has a standard basis of matrix units

$$\{E_{I,J}\}_{I \in \Lambda[n]^l, J \in \Lambda[n]^k} \tag{20}$$

where $E_{I,J}$ has a 1 in the $(I, J)$ position and is 0 elsewhere, we see that, for any standard basis element $e_P \in \Lambda^k(\mathbb{R}^n)$,

$$E_{I,J}e_P = \delta_{J,P}e_I \tag{21}$$

Hence, for any linear map $f : \Lambda^k(\mathbb{R}^n) \to \Lambda^l(\mathbb{R}^n)$, expressing $f$ in the basis of matrix units as

$$f = \sum_{I \in \Lambda[n]^l} \sum_{J \in \Lambda[n]^k} f_{I,J}E_{I,J} \tag{22}$$

we get that

$$f(e_P) = \sum_{I \in \Lambda[n]^l} f_{I,P}e_I \tag{23}$$

By Definition 3.3, $f$ is an $S_n$-equivariant linear map if and only if, for all $\sigma \in S_n$ and standard basis elements $e_J \in \Lambda^k(\mathbb{R}^n)$,

$$f(\rho_k^\Lambda(\sigma)[e_J]) = \rho_l^\Lambda(\sigma)[f(e_J)] \tag{24}$$

Consequently, (24) holds if and only if

$$\sum_{I \in \Lambda[n]^l} \mathrm{sgn}(\pi_{\sigma(J),k})f_{I,\sigma(J)\pi_{\sigma(J),k}}e_I = \sum_{I \in \Lambda[n]^l} f_{I,J} \,\mathrm{sgn}(\pi_{\sigma(I),l})e_{\sigma(I)\pi_{\sigma(I),l}} \tag{25}$$

for some $\pi_{\sigma(J),k} \in S_k$ and $\pi_{\sigma(I),l} \in S_l$ that restores the lexicographical ordering in $\sigma(J)$ and $\sigma(I)$ respectively.

Hence, (25) holds if and only if

$$\mathrm{sgn}(\pi_{\sigma(J),k})f_{\sigma(I)\pi_{\sigma(I),l},\sigma(J)\pi_{\sigma(J),k}} = f_{I,J} \,\mathrm{sgn}(\pi_{\sigma(I),l}) \tag{26}$$

for all $\sigma \in S_n$, $I \in \Lambda[n]^l$ and $J \in \Lambda[n]^k$.

Since $\mathrm{sgn}(\pi_{\sigma(I),l})$ equals $\pm 1$, we see that (26) is equivalent to

$$f_{I,J} = \mathrm{sgn}(\pi_{\sigma(I),l}) \,\mathrm{sgn}(\pi_{\sigma(J),k})f_{\sigma(I)\pi_{\sigma(I),l},\sigma(J)\pi_{\sigma(J),k}} \tag{27}$$

for all $\sigma \in S_n$, $I \in \Lambda[n]^l$ and $J \in \Lambda[n]^k$. $\qquad\square$

*Proof of Lemma 4.12.* Suppose that we are given the normal form of a $(k, l)$–antispherical bipartition having some $t$ $[1, 0]$ blocks where $t \geq 2$ ($t \leq n$ by design already). Fix a set of $t$ distinct labels from $[n]$ for them, and fix the labels of the other spiders in the diagram completely. Then there are $t!$ possible tuples with which to label the $[1, 0]$ spiders in the diagram (for each fixed choice of labels for the other spiders).

However, these $t!$ tuples are in bijective correspondence with the permutations of $S_t$, and so once we reorder the $[1, 0]$ spiders in each of the $t!$ diagrams to be in increasing order, these $t!$ diagrams will all be the same, except half of them will come with positive sign and half will come with negative sign, by the rules in Figure 2. Hence, once we reorder the spiders entirely, half of the reordered diagrams will come with positive sign and half will come with negative sign. This means that the resulting $(I, J)$–entry (corresponding to the $\binom{I}{J}$ element that comes from propagating the block labels to the ends of the wires) in the matrix will be zero.

Since this is true for all possible sets of distinct labels for the $[1, 0]$ spiders and all possible labels for the other spiders, this implies that every $(I, J)$–entry in the matrix is zero.

The proof is exactly the same for $(k, l)$–antispherical bipartitions having some $t$ $[0, 1]$ blocks where $t \geq 2$. $\qquad\square$

*Proof of Theorem 4.13.* Equation (6) tells us that the basis matrices of $\mathrm{Hom}_{S_n}(\Lambda^k(\mathbb{R}^n), \Lambda^l(\mathbb{R}^n))$ can only come from the orbits of the action of $S_n$ on the set $\Lambda[n]^l \times \Lambda[n]^k$. Proposition 4.9 states that these orbits correspond bijectively with the $(k, l)$–antispherical bipartitions having at most $n$ blocks. Combining these statements, we see that the basis matrices of $\mathrm{Hom}_{S_n}(\Lambda^k(\mathbb{R}^n), \Lambda^l(\mathbb{R}^n))$ can only come from the $(k, l)$–antispherical bipartitions having at most $n$ blocks.

Furthermore, Lemma 4.12 showed that any such antispherical bipartition that had more than $[1, 0]$ block or more than $[0, 1]$ block generates the zero matrix. Hence we are left with considering the set of $(k, l)$–antispherical bipartitions having at most $n$ blocks such that each antispherical bipartition has at most $[1, 0]$ block and at most $[0, 1]$ block.

Since equation (6) tells us that if two distinct orbits create non-zero matrices, then these matrices must be different, we just need to show that the remaining $(k, l)$–antispherical bipartitions do in fact create non-zero matrices in order to prove our result.

We can split these into four separate cases. (Note that for certain values of $k$, $l$ and $n$, some of these cases do not appear, but we can approach this with a level of generality).

**Case 1: there are no $[1, 0]$ blocks and no $[0, 1]$ blocks.**

In this case, $k = l$, since the only blocks that appear are $[1, 1]$ blocks. Hence there are $k$ blocks in the antispherical bipartition. Hence we consider all possible $k$-length tuples with distinct elements from $[n]$. We can split this set of tuples into separate classes, where $(a_1, a_2, \ldots, a_k)$ and $(b_1, b_2, \ldots, b_k)$ are in the same class if and only if the elements in the tuples are the same up to some permutation. For each such class, the diagrams that come from labelling the normal form with the tuples in the class all generate the same $\binom{I}{J}$ element once the spiders are reordered, and all with the same sign, since, by Figure 2, swapping two $[1, 1]$ spiders does not change their sign. Hence, for each such class, the $(I, J)$–entry of the corresponding matrix is $+1$. Hence the matrix is non-zero in this case.

**Case 2: there is one $[1, 0]$ block and no $[0, 1]$ blocks.**

In this case, $k = l + 1$, since we must have $l$ $[1, 1]$ blocks and one $[1, 0]$ block. Hence there are $k$ blocks in the antispherical bipartition. As before, we can consider all possible $k$-length tuples with distinct elements from $[n]$. Once again, we split this set of tuples into separate classes, except this time $(a_1, a_2, \ldots, a_k)$ and $(b_1, b_2, \ldots, b_k)$ are in the same class if and only if $a_k = b_k$ and the remaining $k - 1$ elements in each tuple are the same up to some permutation. From here the proof is exactly the same as in Case 1: for each class, we obtain the set of diagrams that come from labelling the normal form with the tuples in the class. For each diagram, we can reorder the $[1, 1]$ spiders in each diagram to be in increasing order without changing the sign, giving us the same diagram overall (repeated many times). Hence once we swap the $[1, 0]$ spider into position, changing sign as appropriate, all of these diagrams generate the same $\binom{I}{J}$ element, all with the same sign. Hence, as before, the matrix is non-zero in this case.

**Case 3: there is no $[1, 0]$ block and one $[0, 1]$ block.**

This is exactly the same as Case 2, except now $l = k + 1$ and there are $k$ $[1, 1]$ blocks and one $[0, 1]$ block. Replacing $k$ by $l$ in Case 2 and $[1, 0]$ by $[0, 1]$ and following exactly the same argument gives the result. Consequently, the resulting matrix is non-zero in this case.

**Case 4: there is one $[1, 0]$ block and one $[0, 1]$ block.**

The final case is also very similar to the previous ones. Here we must have $k = l$, since there are $k - 1 (= l - 1)$ $[1, 1]$ blocks, one $[1, 0]$ block and one $[0, 1]$ block. Hence there are now $k + 1$ blocks in the antispherical bipartition. As before, we can consider all possible $(k + 1)$-length tuples with distinct elements from $[n]$. Once again, we split this set of tuples into separate classes, except this time $(a_1, a_2, \ldots, a_k, a_{k+1})$ and $(b_1, b_2, \ldots, b_k, b_{k+1})$ are in the same class if and only if $a_k = b_k$, $a_{k+1} = b_{k+1}$ and the remaining $k - 1$ elements in each tuple are the same up to some permutation. We now follow the same argument as before: for each class, we obtain the set of diagrams that come from labelling the normal form with the tuples in the class. For each diagram, we can reorder the $[1, 1]$ spiders in each diagram to be in increasing order without changing the sign, giving us the same diagram overall (repeated many times). Hence once we swap the $[1, 0]$ spider and the $[0, 1]$ spider into position, changing sign as appropriate, all of these diagrams generate the same $\binom{I}{J}$ element, all with the same sign. Hence, as before, the matrix is non-zero in this case. $\square$

## B    EMBEDDING INTO TENSOR POWER SPACE

In Section 5, we said that we would like to be able to represent any $S_n$-equivariant weight matrix $W$ from $\Lambda^k(\mathbb{R}^n)$ to $\Lambda^l(\mathbb{R}^n)$ as a matrix from $(\mathbb{R}^n)^{\otimes k}$ to $(\mathbb{R}^n)^{\otimes l}$ so that it can be applied to antisymmetric tensors in $(\mathbb{R}^n)^{\otimes k}$. We call the resulting matrix from $(\mathbb{R}^n)^{\otimes k}$ to $(\mathbb{R}^n)^{\otimes l}$ the **unrolled** $S_n$-equivariant weight matrix, and the process of representing the original matrix in this way **unrolling**.

It is easiest to perform unrolling on each individual basis matrix. Since each non-zero entry in a rolled basis matrix corresponds to a labelled, reordered antispherical bipartition diagram with the corresponding sign attached, we obtain the entries in the unrolled basis matrix by permuting the spiders in this diagram into all possible orders, again tracking sign changes. For each resulting order, we propagate the block labels to the end of each wire, obtaining some $\binom{I}{J}$ element with a sign attached. These then give the non-zero $(I, J)$–entries in the unrolled basis matrix (positive sign is $+1$, negative sign is $-1$), with all other entries being zero. We can then weight each unrolled basis matrix as before and add them together to obtain the unrolled weight matrix.

This holds because the $k^{\text{th}}$ antisymmetric power of $\mathbb{R}^n$ can be thought of as the quotient space of $(\mathbb{R}^n)^{\otimes k}$ by the ideal generated by elements of the form $x \otimes y + y \otimes x$.

*Example* B.1. Returning to Example 4.16, we see that the unrolled $S_3$-equivariant weight matrix from $(\mathbb{R}^3)^{\otimes 2}$ to $\mathbb{R}^3$ is

$$
\begin{array}{c}
\begin{array}{ccccccccc}
\color{blue}1,1 & \color{blue}1,2 & \color{blue}1,3 & \color{blue}2,1 & \color{blue}2,2 & \color{blue}2,3 & \color{blue}3,1 & \color{blue}3,2 & \color{blue}3,3
\end{array} \\
\begin{array}{c}\color{blue}1\\\color{blue}2\\\color{blue}3\end{array}
\left[\begin{array}{ccccccccc}
0 & w & w & -w & 0 & 0 & -w & 0 & 0 \\
0 & -w & 0 & w & 0 & w & 0 & -w & 0 \\
0 & 0 & -w & 0 & 0 & -w & w & w & 0
\end{array}\right]
\end{array}
\tag{28}
$$

We provide the calculations in Example D.2.

## C    ALGORITHM TO GENERATE ALL $(k, l)$–ANTISPHERICAL BIPARTITIONS

Since we know that the total number of wires in a $(k, l)$–antispherical bipartition is $k + l$, we can generate all possible $(k, l)$–antispherical bipartitions by considering certain colourings of a tableau with $l$ boxes in the first row and $k$ boxes in the second row.

For example, if $l = 2$ and $k = 3$, then the tableau has the shape

$$\tag{29}$$

We only allow three colours, red, blue, and green, such that

- if a box in the first row is coloured in red, then there must be a box directly below it that is also coloured in red,

- if a box in the second row is coloured in red, then there must be a box directly above it that is also coloured in red,

- if a box is coloured in green, then it must only appear in the first row, and

- if a box is coloured in blue, then it must only appear in the second row.

Moreover, since we want to generate the normal form of a $(k, l)$–antispherical bipartition, we can only colour boxes from the left in red boxes from the right in blue or green.

From here, we are able to construct such an algorithm.

First, we take the tableau and initially colour it as follows:

- if $k > l$, then we colour the last $k - l$ boxes in the second row with blue, and the remaining boxes with red,

- if $k < l$, then we colour the last $l - k$ boxes in the first row with green, and the remaining boxes with red,

- otherwise, we colour all boxes with red.

For example, we would obtain the following initial tableaux for different values of $k$ and $l$:

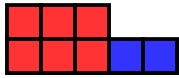 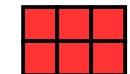 (30)

Then, starting from the right, we change the colours of each red column by colouring the box in the first row with green and the box in the second row with blue, creating new coloured tableau as we move towards the left, until we form a tableau that has only blue and green colours.

For example, for the initial tableau on the left in (30), this process generates the following tableaux:

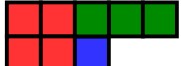 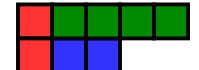 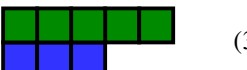 (31)

Each of these tableau corresponds to a $(k, l)$–antispherical bipartition as follows:

- each column consisting of two red boxes corresponds to a $[1, 1]$ block,
- each blue box corresponds to a $[1, 0]$ block, and
- each green box corresponds to a $[0, 1]$ block.

i.e.

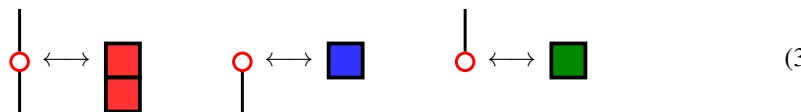 (32)

For example, we see that the left most diagram in (31) corresponds to the $(3, 5)$–antispherical bipartition

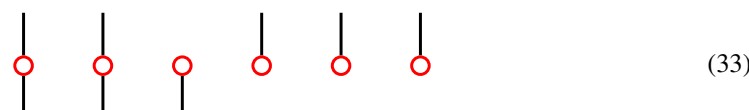 (33)

Clearly, this algorithm generates all possible $(k, l)$–antispherical bipartitions. In fact, we know their total number.

**Lemma C.1.** *The number of $(k, l)$–antispherical bipartitions is $1 + \min(k, l)$.*

*Proof.* This is immediate from the construction. We initially generate a tableau with $\min(k, l)$ columns, all coloured red, and depending on the values of $k$ and $l$, potentially an additional right tail of cells coloured in either green or blue. From there we generate new tableaux by changing each of the $\min(k, l)$ columns in this tableau, one by one, from two red colours to a column consisting of a green and blue coloured cell. Since each tableau corresponds to a $(k, l)$–antispherical bipartition, there are $1 + \min(k, l)$ such antispherical bipartitions. □

We would also like to generate the subset of $(k, l)$–antispherical bipartitions having at most $n$ blocks with at most one $[1, 0]$ block and at most one $[0, 1]$ block. With the latter restriction, the algorithm given above shows that:

- if $k$ and $l$ differ by 2 or more, then there are no such antispherical bipartitions;
- if $k$ and $l$ differ by 1, then there is only one such antispherical bipartition so long as $n \geq \max(k, l)$, otherwise there are none; and
- if $k$ and $l$ are equal, then there are
  - two such antispherical bipartitions if $n > k \ (= l)$,
  - one such antispherical bipartition if $n = k \ (= l)$,
  - no such antispherical bipartitions if $n < k \ (= l)$,

Since these $(k, l)$–antispherical bipartitions correspond bijectively to the basis matrices of $\mathrm{Hom}_{S_n}(\Lambda^k(\mathbb{R}^n), \Lambda^l(\mathbb{R}^n))$, by Theorem 4.13, the above characterisation immediately gives us its dimension.

## D ADDITIONAL EXAMPLES

*Example* D.1 (Calculations for Example 4.16). We first need to consider all possible 2-length tuples with elements in $[3]$, not allowing for repetitions amongst the elements, and use those to label the central red nodes of the normal form of the $(2, 1)$–antispherical bipartition $\{[1, 1], [1, 0]\}$. They give six labelled diagrams, namely:

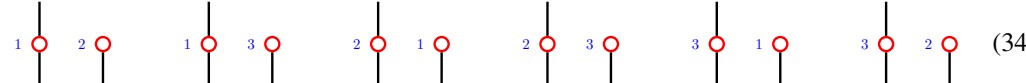

$$(34)$$

Next, we reorder the spiders in each diagram so that they are in increasing order from left to right, tracking any sign changes as appropriate.

$$(35)$$

We propagate the block labels to the end of each wire to produce the element $\binom{I}{J}$ with accompanying sign:

$$\binom{1}{12} : +1 \quad \binom{1}{13} : +1 \quad \binom{2}{12} : -1 \quad \binom{2}{23} : +1 \quad \binom{3}{13} : -1 \quad \binom{3}{23} : -1 \tag{36}$$

Hence we see that the basis matrix that corresponds to this antispherical bipartition is

$$\begin{array}{c} \\ 1 \\ 2 \\ 3 \end{array} \begin{array}{ccc} {\scriptstyle 1,2} & {\scriptstyle 1,3} & {\scriptstyle 2,3} \\ \begin{bmatrix} 1 & 1 & 0 \\ -1 & 0 & 1 \\ 0 & -1 & -1 \end{bmatrix} \end{array} \tag{37}$$

*Example* D.2 (Calculations for Example B.1). We would now like to unroll the weight matrix that was found in Example D.1. We follow the steps that were given in Section B.

We take each of the labelled, reordered antispherical bipartition diagrams that were given in (35) — since each one corresponds to a non-zero entry in the weight matrix (37) — and calculate all possible orders of the spiders for each one, tracking sign changes. This gives us

$$(38)$$

We propagate the block labels to the end of each wire to produce the element $\binom{I}{J}$ with accompanying sign:

$$\binom{1}{21} : -1 \quad \binom{1}{31} : -1 \quad \binom{2}{21} : +1 \quad \binom{2}{32} : -1 \quad \binom{3}{31} : +1 \quad \binom{3}{32} : +1 \tag{39}$$

Hence, by taking the $\binom{I}{J}$ elements with their accompanying sign from both (36) and (39), we see that the unrolled basis matrix that corresponds to the antispherical bipartition $\{[1, 1], [1, 0]\}$ is

$$\begin{array}{c} \\ 1 \\ 2 \\ 3 \end{array} \begin{array}{ccccccccc} {\scriptstyle 1,1} & {\scriptstyle 1,2} & {\scriptstyle 1,3} & {\scriptstyle 2,1} & {\scriptstyle 2,2} & {\scriptstyle 2,3} & {\scriptstyle 3,1} & {\scriptstyle 3,2} & {\scriptstyle 3,3} \\ \begin{bmatrix} 0 & 1 & 1 & -1 & 0 & 0 & -1 & 0 & 0 \\ 0 & -1 & 0 & 1 & 0 & 1 & 0 & -1 & 0 \\ 0 & 0 & -1 & 0 & 0 & -1 & 1 & 1 & 0 \end{bmatrix} \end{array} \tag{40}$$

*Example* D.3 (Deep Sets $S_n$-Equivariant Weight Matrix from $\mathbb{R}^n$ to $\mathbb{R}^n$). To find the Deep Sets $S_n$-equivariant weight matrix from from $\mathbb{R}^n$ to $\mathbb{R}^n$, we need to find a basis of $\mathrm{Hom}_{S_n}(\Lambda^1(\mathbb{R}^n), \Lambda^1(\mathbb{R}^n))$. since $\Lambda^1(\mathbb{R}^n) = \mathbb{R}^n$.

We need to consider all $(1, 1)$–antispherical bipartitions having at most $n$ blocks. Assuming that $n > 1$, they are

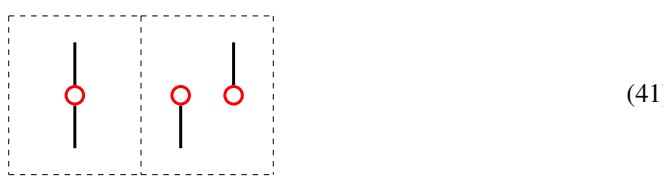

$$(41)$$

For the first diagram in (41), the only valid 1-length tuples are the elements of $[n]$. Hence, the matrix that corresponds to this $(1,1)$–antispherical bipartition is the $n \times n$ identity matrix.

For the second diagram in (41), all possible 2-length tuples with distinct elements from $[n]$ are valid. Since we can swap the spiders around in this diagram without changing sign, by the rules that are given in Figure 2, this means that the matrix that corresponds to this antispherical bipartition is the $n \times n$ matrix that is 0 on the diagonal and 1 elsewhere.

Hence, the $(i,j)$-entry of the $S_n$-equivariant weight matrix from $\mathbb{R}^n$ to $\mathbb{R}^n$ is $w_1$ if $i = j$ and $w_2$ if $i \neq j$, for weights $w_1, w_2 \in \mathbb{R}$. This recovers the Deep Sets characterisation that first appeared in Zaheer et al. (2017).

*Example* D.4. We said in Remark 5.3 that as soon as we obtained the IO pattern for a $(k,l)$–antispherical bipartition then we could use it to generate the basis matrix for any $n$ that is greater than or equal to the number of blocks in the antispherical bipartition.

We saw in Example 5.2 that the IO pattern that corresponds to the $(2,1)$–antispherical bipartition $\{[1,1],[1,0]\}$ is

$$\longleftrightarrow \quad i \leftarrow \sum_{j:j\neq i}^{n} i,j - \sum_{j:j\neq i}^{n} j,i \tag{42}$$

Hence, for example, the unrolled $S_4$-equivariant weight matrix from $(\mathbb{R}^4)^{\otimes 2}$ to $\mathbb{R}^4$ is

$$
\begin{array}{c c}
 & \begin{array}{cccccccccccccccc} 1,1 & 1,2 & 1,3 & 1,4 & 2,1 & 2,2 & 2,3 & 2,4 & 3,1 & 3,2 & 3,3 & 3,4 & 4,1 & 4,2 & 4,3 & 4,4 \end{array} \\
\begin{array}{c} 1 \\ 2 \\ 3 \\ 4 \end{array} & \left[\begin{array}{cccccccccccccccc}
0 & w & w & w & -w & 0 & 0 & 0 & -w & 0 & 0 & 0 & -w & 0 & 0 & 0 \\
0 & -w & 0 & 0 & w & 0 & w & w & 0 & -w & 0 & 0 & 0 & -w & 0 & 0 \\
0 & 0 & -w & 0 & 0 & 0 & -w & 0 & w & w & 0 & w & 0 & 0 & -w & 0 \\
0 & 0 & 0 & -w & 0 & 0 & 0 & -w & 0 & 0 & 0 & -w & w & w & w & 0
\end{array}\right]
\end{array} \tag{43}
$$

since there is only one basis matrix that appears in its construction.

**A General Procedure for Calculating the $S_n$-Equivariant Weight Matrix from Antisymmetric Tensors in $(\mathbb{R}^n)^{\otimes k}$ to Antisymmetric Tensors in $(\mathbb{R}^n)^{\otimes l}$.**

Perform the following steps:

1. Calculate all of the $(k,l)$–antispherical bipartitions having at most $n$ blocks such that each antispherical bipartition has at most one $[1,0]$ block and at most $[0,1]$ block.

2. For each $(k,l)$–antispherical bipartition, expressed in its normal form, calculate:
   - For a generic fixed index $I = (i_1, i_2, \ldots, i_l)$, all of the possible ways in which we can label the spiders together with the permutations of the spiders for a given labelling that create new inputs which map to this fixed output.
   - This produces an IO pattern which has $I$ on the left hand side of the arrow $\leftarrow$ and the sum of all of these inputs, each with its sign that comes from the labelled diagram, on the right hand side of $\leftarrow$, where indices on the right hand side that do not appear in $I$ are also summed over.
   - Forming an $n^l \times n^k$ matrix, for each tuple $I$ consisting of distinct elements from $[n]$, and for each tuple $J$ that appears on the right hand side of the IO pattern for this choice of $I$, we place in the $(I,J)$–entry of the matrix $+1$ if the sign for $J$ in the IO pattern is positive and $-1$ if the sign is negative.

3. The resulting matrices are the basis matrices. Attaching a weight $\lambda_\pi \in \mathbb{R}$ to each and adding them together gives the overall unrolled weight matrix.

In reality, we have shown that we do not need to explicitly calculate the weight matrix in order to be able to implement the transformation: it is enough to use the IO pattern that comes from each of the $(k,l)$–antispherical bipartitions described above.

# E  LIMITATIONS

This work is primarily theoretical, providing a complete characterisation of linear permutation equivariant functions between antisymmetric tensors. While its immediate practical impact may be limited, we believe that it lays a principled foundation for future developments in neural network architectures and learning algorithms that are capable of operating on data that can be represented by antisymmetric tensors, with potential applications in real-world domains where these structures naturally arise.

# F  REFERENCE LIST: IO PATTERNS FOR $(k, l)$–ANTISPHERICAL BIPARTITIONS

We provide some IO patterns for commonly appearing $(k, l)$–antispherical bipartitions below. Note that the IO pattern only appears for $n$ greater than the number of spiders in the antispherical bipartition.

| $(k, l)$–Antispherical Bipartition | IO Pattern |
|---|---|
|  | $i \leftarrow i$ |
|  | $i, j \leftarrow i, j$ |
|  | $i \leftarrow \sum_{j: j \neq i}^{n} [i, j - j, i]$ |
|  | $i, j \leftarrow i - j$ |
|  | $i \leftarrow \sum_{j: j \neq i}^{n} j$ |
|  | $i, j \leftarrow \sum_{k: k \neq i, j}^{n} [i, j, k - i, k, j + k, i, j]$ |
|  | $i, j, k \leftarrow i, j - i, k + j, k$ |
|  | $i, j \leftarrow \sum_{k: k \neq i, j}^{n} [i, k - k, i - j, k + k, j]$ |

# G    IMPLEMENTATION DETAILS

We describe the architecture and training details that were used for each of the two tasks. In implementing the linear permutation equivariant functions, we used batch vectorised operations for each of the unrolled basis transformations that came from the corresponding IO patterns. The models were implemented in PyTorch 2.2.2 and the experiments were run on a MacBook Pro with an Apple M1 Max CPU (10 cores) and 64 GB of RAM.

$S_8$-**Equivariant Task:**    We randomly generated a synthetic data set that was composed of 10000 antisymmetric tensors in the training set and 2000 antisymmetric tensors in the test set. We used a manual seed of 42 for both the data generation process and the training so that the results are reproducible. We trained three models which had the following specifications:

- AntiSymmPermEquiv, the $S_8$-equivariant model on antisymmetric tensors, consisted of an $S_8$-equivariant linear layer between antisymmetric tensors in $(\mathbb{R}^8)^{\otimes 2}$ and $\mathbb{R}^8$ respectively, then a ReLU activation function, followed by two layers each consisting of an $S_8$-equivariant linear layer (between antisymmetric tensors) from $\mathbb{R}^8$ to $\mathbb{R}^8$ and a ReLU activation function. The total number of parameters was 5 and the training time was 4.67 seconds.

- PermEquiv, the standard $S_8$-equivariant model (Maron et al., 2019a; Pearce-Crump, 2024; Godfrey et al., 2023), consisted of an $S_8$-equivariant linear layer between tensors in $(\mathbb{R}^8)^{\otimes 2}$ and $\mathbb{R}^8$ respectively, then a ReLU activation function, followed by two layers each consisting of an $S_8$-equivariant linear layer from $\mathbb{R}^8$ to $\mathbb{R}^8$ and a ReLU activation function. The total number of parameters was 9 and the training time was 5.52 seconds.

- MLP consisted of a standard $\mathbb{R}^{8 \times 8}$ to $\mathbb{R}^8$ linear layer, then a ReLU activation function, followed by two layers each consisting of a standard $\mathbb{R}^8$ to $\mathbb{R}^8$ linear layer and a ReLU activation function. The total number of parameters was 664 and the training time was 3.56 seconds.

All three models were optimised with stochastic gradient descent with a learning rate of 0.001. We trained the models for 20 epochs with a batch size of 32 which resulted in the training losses (mean squared error (MSE)) that appeared in Figure 3. Given that the models were relatively small, memory usage was not a concern, and they comfortably fit within the available 64 GB of RAM.

$S_4$-**Invariant Task:**    We randomly generated synthetic data sets composed of 30, 300, 3000, and 30000 antisymmetric tensors in the training set, and a test set composed of 1000 antisymmetric tensors. The synthetic data was generated with a random seed, which was incremented for each run (five runs in total) to ensure different data splits and variability in the results. The base seed value was set to 42 so that the results are reproducible. We trained three models which had the following specifications:

- AntiSymmPermEquiv, the $S_4$-invariant model on antisymmetric tensors, consisted of an $S_4$-equivariant linear layer between antisymmetric tensors in $(\mathbb{R}^4)^{\otimes 2}$ and $\mathbb{R}^4$ respectively, then a ReLU activation function, then an $S_4$-equivariant linear layer between (antisymmetric) tensors in $\mathbb{R}^4$ and $\mathbb{R}^4$ respectively, then a ReLU activation function, followed by an $S_4$-invariant linear layer from $\mathbb{R}^4$ to $\mathbb{R}$ and a ReLU activation function. The total number of parameters was 4.

- PermEquiv, the standard $S_4$-invariant model (Maron et al., 2019a; Pearce-Crump, 2024; Godfrey et al., 2023), consisted of an $S_4$-equivariant linear layer between tensors in $(\mathbb{R}^4)^{\otimes 2}$ and $\mathbb{R}^4$ respectively, then a ReLU activation function, followed by the same layers that appeared in AntiSymmPermEquiv. The total number of parameters was 8.

- MLP consisted of a standard $\mathbb{R}^{4 \times 4}$ to $\mathbb{R}^4$ linear layer, with a ReLU activation function, then a standard $\mathbb{R}^4$ to $\mathbb{R}^4$ linear layer with ReLU, followed by a standard $\mathbb{R}^4$ to $\mathbb{R}$ linear layer with ReLU. The total number of parameters was 93.

All three models were optimised with stochastic gradient descent with a learning rate of 0.001. We trained the models for 20 epochs with a batch size of 32 which gave the training losses (mean squared error (MSE)) that appeared in Table 2. Given that the models were relatively small, memory usage was not a concern, and they comfortably fit within the available 64 GB of RAM.

