# OpenReview forum: "Permutation Equivariant Neural Networks for Antisymmetric Tensors"
_ICLR.cc/2026/Conference — Submitted to ICLR 2026_

### Official Review · Reviewer_DvEj · 2025-10-27

**Soundness:** 3
**Presentation:** 3
**Contribution:** 2
**Rating:** 4
**Confidence:** 5

**Summary:**

The paper introduces permutation-equivariant models for antisymmetric tensors, i.e., for the antisymmetric powers of $\mathbb{R}^n$.

**Strengths:**

- I find the theoretical contribution to be **the strongest aspect of the paper**. Sections 3 and 4 are particularly enjoyable for readers interested in theory, as they provide a complete characterization of linear permutation-equivariant scalar-valued (and vector-valued, respectively) functions on antisymmetric tensors in $(\mathbb{R}^n)^{\otimes k}$ as elements of $\mathrm{Hom}{S_n}(\Lambda^k(\mathbb{R}^n), \Lambda^0(\mathbb{R}^n))$ (and $\mathrm{Hom}{S_n}(\Lambda^k(\mathbb{R}^n), \Lambda^1(\mathbb{R}^n))$, respectively). The authors also supply all necessary proofs in the Appendix. A solid background in representation theory would likely help readers fully appreciate the results.

- This theoretical foundation naturally leads to the efficient implementation of the proposed layers in Section 5, which is then applied in the experimental results presented in Section 6. While I am able to follow the high-level ideas in Section 5, I do not feel fully confident evaluating the correctness of the implementation details, as I lack a strong background in coding. However, I did notice that imposing equivariance appears to constrain many parameters in the linear layers--forcing some to be zero or equal to one another--which may substantially reduce the expressiveness of the resulting model.

**Weaknesses:**

I believe the empirical aspect of the paper is its weakest part.

- Both experiments are conducted on very small synthetic datasets, which makes it difficult to draw meaningful conclusions about the practical effectiveness or generalizability of the proposed approach.

- As mentioned earlier, the equivariance constraints drastically reduce the number of learnable parameters--down to only 5 in the first experiment and 4 in the second. Even the next-best baseline models are extremely small (9 and 8 parameters, respectively). In contrast, the unconstrained MLPs, while still modest in size (664 and 93 parameters), have significantly more freedom to learn.

Although I understand that the symmetry constraints inherently shrink the parameter space, experiments on such tiny models and toy datasets provide limited insight. They do not convincingly demonstrate whether the proposed architectures can scale, whether they can handle realistic noise and variability.

**Questions:**

To strengthen the empirical section, I would encourage the authors to include at least one experiment on a real dataset, and a significantly larger synthetic dataset, together with a scaling analysis illustrating how the method behaves as model size and data complexity increase. Without such evidence, the experiments read more as a proof-of-concept demonstration rather than a convincing empirical validation.

---

My current score reflects the fact that, while I acknowledge and appreciate the impressive theoretical contribution of the paper, I believe that for a conference focused on Machine Learning, the empirical aspect also plays an important role. In its present form, the work feels more suitable for a theory-oriented venue, such as a journal, where empirical validation is typically less central to the evaluation criteria.

---

> ### Author Response · Authors · 2025-11-17
>
> We thank the reviewer for their thoughtful comments. We are pleased that they appreciate “the impressive theoretical contribution” of our paper and recognise that it contains “all necessary proofs in the Appendix”.
>
> Although we designed our experiments to be proof-of-concept to reflect the fact that our paper’s primary contribution is theoretical, we appreciate the reviewer’s suggestion that providing stronger empirical evidence is important for the ICLR community. In response to all of the feedback from all of the reviewers, we have conducted 1) an experiment on a real-world application (with a real-world dataset) and 2) a scaling and noise analysis on a significantly larger synthetic dataset. We detail these below:
>
> **Experiment on real-world application**
>
> We evaluated our model on the Wikipedia Requests for Adminship (Wiki-RfA) dataset (https://snap.stanford.edu/data/wiki-RfA.html), where nodes represent Wikipedia members and directed edges correspond to votes, with positive values indicating support for adminship and negative values indicating opposition. We filtered the dataset for neutral votes to retain only directional interactions, resulting in a directed graph containing 332 nodes and 243 edges.
>
> We sampled 6-node subgraphs (80 training, 20 test) using a fixed random seed to preserve the local structure and ensure variability for training.  Each subgraph was represented by its antisymmetric adjacency matrix, which was adjusted by a small Gaussian noise to improve robustness and reduce potential overfitting. For every subgraph, we constructed a node-level target by multiplying with the adjacency matrix to accumulate incoming influence over 1 to 3 hops, down-weighting each hop by a decay factor so that direct votes contributed most and indirect influence diminished with distance.
>
> The task was to estimate a user’s local influence within a subgraph. Over five runs, the antisymmetric model consistently outperformed a standard MLP baseline (Test MSE: 0.039 ± 0.018 vs. 0.152 ± 0.029), showing that explicitly modelling the antisymmetry allows the model to learn the local influence propagation more accurately than an unconstrained MLP, even though the MLP has more parameters. This experiment provides evidence of our approach’s applicability to real-world settings, directly addressing the reviewers’ concerns about its relevance in a practical setting.
>
> [Model and additional training details. We compared our model against a baseline MLP. Our model consisted of six layers of our permutation equivariant linear maps on antisymmetric tensors from 2->1, 2->2, 1->2, and 1->1 (our layer maps on different orders) together with tanh activation functions. Each subgraph, represented as a 6×6 antisymmetric adjacency matrix, served as input. The baseline MLP used a comparable multi-layer structure but without the permutation equivariance or the antisymmetric structure. We trained both models over three independent runs for 100 epochs using the Adam optimizer with a learning rate of 0.01 and mean squared error as the loss function.]

---

> > ### Author Response · Authors · 2025-11-17
> >
> > **Scaling and noise analysis on a significantly larger synthetic dataset**
> >
> > For this experiment, we generated synthetic graphs of varying sizes (10, 20, 40, 80, and 160 nodes), where each graph was constructed as a directed Erdos-Renyi network with edge probability 0.3, which we then converted into an antisymmetric adjacency matrix. For each graph in a given size, we added noise that was normally distributed with zero mean and standard deviations from (0.01, 0.05, 0.1, 0.2, 0.3, 0.4) to evaluate the robustness of our network under noisy data.
> >
> > We defined each node’s target in an analogous way to the real-world experiment: the influence was aggregated along paths of up to three hops, with each hop’s contribution down-weighted by a decay factor. We also randomly dropped a fraction of the paths to introduce some sparsity in the target values. We normalised the resulting vector for each graph by dividing by its maximum absolute value. For each pair of size and noise, we sampled 1000 graphs (split into 800 training graphs and 200 test graphs) and trained the model over five independent runs. Our results are the following (Test MSE - mean ± 1 standard deviation):
> >
> > | Graph Size/Noise | 0.01 | 0.05 | 0.10 | 0.20 | 0.30 | 0.40 |
> > |-----------:|------|------|------|------|------|------|
> > | **10**  | 0.2151 ± 0.0117 | 0.2122 ± 0.0060 | 0.2096 ± 0.0046 | 0.2052 ± 0.0047 | 0.2005 ± 0.0061 | 0.1989 ± 0.0084 |
> > | **20**  | 0.1558 ± 0.0084 | 0.1575 ± 0.0081 | 0.1557 ± 0.0078 | 0.1528 ± 0.0081 | 0.1500 ± 0.0078 | 0.1475 ± 0.0068 |
> > | **40**  | 0.1664 ± 0.0359 | 0.1618 ± 0.0328 | 0.1587 ± 0.0310 | 0.1579 ± 0.0269 | 0.1520 ± 0.0216 | 0.1517 ± 0.0196 |
> > | **80**  | 0.1361 ± 0.0258 | 0.1364 ± 0.0272 | 0.1358 ± 0.0282 | 0.1366 ± 0.0274 | 0.1359 ± 0.0271 | 0.1339 ± 0.0258 |
> > | **160** | 1.6170 ± 2.9674 | 1.1715 ± 2.0801 | 0.9946 ± 1.7268 | 1.2886 ± 2.3151 | 1.0306 ± 1.7993 | 1.4292 ± 2.5962 |
> >
> > The results show that the model performs well on small-sized and medium-sized graphs across all noise levels, with higher loss and its standard deviation for the largest graphs (160 nodes), reflecting the greater difficulty of predicting their targets. Overall, these results demonstrate our model’s robustness to noise and its predictable scaling behaviour as the graph size increases, directly addressing the reviewer’s request for such analysis on a larger synthetic dataset.
> >
> > Finally, the reduced number of parameters in our layers arises directly from encoding the structural and symmetry constraints in them. This is an important feature of our networks rather than a limitation: by incorporating the relevant symmetries, our layers are able to learn target functions that unconstrained MLPs cannot, despite them having more parameters. We agree that understanding expressiveness under these constraints is important, and we see this as a natural direction for future work.

---

> > > ### Comment · Reviewer_DvEj · 2025-11-26
> > >
> > > I thank the Authors for the detailed response.
> > >
> > > As stated in my original review, the theoretical contribution of the paper is strong and well-developed, but the empirical component was initially insufficient. I appreciate the authors’ effort in responding to these concerns. The newly added experiments substantially strengthen the submission: the real-world Wiki-RfA evaluation demonstrates that the antisymmetric model can outperform a standard MLP even with far fewer parameters, suggesting that the imposed symmetry can be beneficial rather than restrictive. The scaling and noise study up to 160-node graphs further provides useful evidence of robustness and predictable behavior as graph size increases. The clear reporting of MSE statistics, multiple independent runs, and methodological details improves the reproducibility and interpretability of the results. (A minor point: the variance for graph size 160 appears quite high relative to the mean.)
> > >
> > > These additions sufficiently address my earlier concerns. I am raising my score to 6, with a positive stance toward the submission. I recommend that the authors incorporate the new experimental results into the revised manuscript, as the venue permits revision.

---

### Official Review · Reviewer_DPLT · 2025-10-31

**Soundness:** 4
**Presentation:** 3
**Contribution:** 3
**Rating:** 8
**Confidence:** 3

**Summary:**

This paper provides a classification of all linear maps from $\wedge^k(\mathbb{R}^n)$ to $\wedge^l(\mathbb{R}^n)$ which are permutation equivariant. This is relevant to deep learning since it is equivalent to providing a full classification of all linear layers that are permutation equivariant and preserve the antisymmetric structure of these spaces (something that does not naturally happen when $\wedge^k(\mathbb{R}^n)$ is interpreted as a product of standard vector spaces during implementation). The paper begins by reviewing basic properties of $\wedge^k(\mathbb{R}^n)$. Then the paper gives its classification of all linear maps in $\hom_{S_n}(\wedge^k(\mathbb{R}^n), \wedge^l(\mathbb{R}^n))$ using some simple diagrammatics and a combinatorial model which the paper calls antispherical bipartitions. Since elements of $\hom_{S_n}(\wedge^k(\mathbb{R}^n), \wedge^l(\mathbb{R}^n))$ can become very large when explicitly manifested, the paper describes an efficient way for encoding these via what the paper calls *$IO$ patterns*. Finally, as a proof-of-concept, the paper presents some small-scale numerical experiments that show that models using the layers described in the paper perform better than naïve methods.

**Strengths:**

**Clarity:** This is a well-written paper, that reads like a math paper (in a good way). Notation is clearly laid out so that no guessing is required on the part of the reviewer. The paper reviews the exterior product and establishes its basic properties before launching into the analysis of $\hom_{S_n}(\wedge^k(\mathbb{R}^n), \wedge^l(\mathbb{R}^n))$. Despite this being in some ways a fairly abstract paper, the writing style makes it easier to read than many more concrete papers that deal with simpler constructions.

**This paper contains elegant mathematics that tells a nice story:** The reviewer appreciated the use of combinatorial and diagrammatic constructions in the paper’s classification of elements of $\hom_{S_n}(\wedge^k(\mathbb{R}^n), \wedge^l(\mathbb{R}^n))$. This style has a long history within the pure math community and to some degree physics, but one rarely sees it in machine learning. In this paper, the approach really brought the results to life and made them much more tangible than would be the case if the analysis was performed in a more brute force style (e.g., complicated descriptions of matrices). As a result of the way the mathematics is developed, a lot of the theorems appear evident, even though they are non-trivial.

**Weaknesses:**

**What is already known and what is new?:** Exterior powers are a classical construction and as such, much is known about their structure, including their Hom spaces. It was unclear to the reviewer what exactly was new here and what is already known and simply being applied to the problem of layers in neural networks. It was at least stated that the antispherical bipartitions are a new construction. Clarifying these points would help the reader better understand how the work fits into the research landscape.

**Further experimental analysis:** The reviewer appreciates that the main point of this work is not experimental and that realistic data with antisymmetric structure (e.g., data coming from physics or differential geometry) is likely to be unhelpful to the general reader. However, there are a range of interesting experiments that could be run with the synthetic data that is already used in the paper. For example, the reviewer would have been interested to see what happened when $n$, $k$, or $l$ are varied. When does learning break down?

**More information on the types of applications where this is relevant:** One limitation of this type of work is that most of the ICLR community probably doesn’t work with data that has an antisymmetric structure. One idea for engaging such a reader would be to describe how this structure arises in an application (preferably the most accessible one available). This would help motivate and ground the rest of the work. The mix of combinatorics and algebra that this work taps into is very beautiful so making it accessible to the broader community would be a benefit to everyone.

**The topic is somewhat niche:** While exterior products and morphisms between them are certainly a fundamental part of mathematics, it is unclear how much the machine learning community needs networks that can accommodate them. The work seems to suggest that so far, few papers actually tackle this problem. It was unclear to the reviewer if this was because there simply isn’t a demand in applications or if there is a demand and practitioners are simply using unsuitable architectures.

**Nitpicks:**
- Line 025: I believe that the history of equivariant networks goes back quite a bit further than this. It is this reviewer’s understanding that this was a central feature in the development of convolutional neural networks for instance.
- Line 157: The reviewer liked the use of this colored box. But it would be useful to put a title on it for later reference.

**Questions:**

- Networks typically contain a few different types of layers, how do layers like non-linearity, norms, etc. interact with this type of equivariance?
- What sizes of $k$, $l$ can we realistically expect to compute with?

---

> ### Author Response · Authors · 2025-11-17
>
> We thank the reviewer for taking the time to review our paper, and for providing such a positive critique. We felt that the reviewer really understood the purpose of our paper and how it fit into the wider landscape of other papers characterising group equivariant linear layers.
>
> In particular, we are glad that the reviewer recognises the “non-trivial” results in our paper, that they enjoyed the approach that we took in our presentation of these results, and that they found them to be “very beautiful”, “well-written” and “easy to read”. We agree with them that the style we took “has a long history within the pure math community and to some degree physics, but one rarely sees it in machine learning”.
>
> The reviewer asks us to clarify the novelty of our results. They are correct that classically much is known about the structure of exterior powers and their Hom spaces. However (and to the best of our knowledge), nothing was previously known about their Hom_{S_n} spaces, that is the S_n equivariant maps, until our paper. This is a central part of our contribution to knowledge. Furthermore, the tool that we use to obtain the characterisation (the antispherical bipartitions) is also a new construction designed to obtain such a characterisation. Finally, the use of what we have termed IO patterns is new in terms of obtaining an improved implementation of these specific layers over standard matrix multiplication, which becomes ever more important regarding memory usage as the values of $n$, $k$, $l$ scale.
>
> We agree that we could do some further interesting analysis on how the model scales. This was also something that was requested by Reviewer DvEj. In our rebuttal to their review, we have provided such an analysis, focusing on how the model scales as the value of $n$ changes, which we think is most valuable given our comments on the likely values of $k$ and $l$ in practical settings (see below in our response to this reviewer’s questions).
>
> The reviewer also makes an excellent suggestion to help engage ICLR readers in our work, namely by asking us to describe how the antisymmetric structure arises in an application. The most accessible one (that we could think of) is determining influence in a social network, where we can obtain an antisymmetric adjacency matrix from the network, viewed as a directed graph. Notably the function that we need to learn in this setting should be permutation equivariant, because the ordering of nodes is arbitrary. We have conducted an experiment on real-world data to study this problem (see our rebuttal to Reviewer DvEj), where our model outperformed an unconstrained MLP. We agree that including this example and the resulting analysis “would help motivate and ground the rest of the work.”
>
> On the question of demand: we believe that the limited amount of prior work on this issue is due to the absence of suitable architectures. By providing networks that respect the antisymmetry in the tensors and the permutation equivariant structure, we enable the exploration of applications where this structure is relevant, such as modelling pairwise interactions in molecules or directed flow networks.
>
> On the nitpicks:
>
> -	We agree that equivariance has been used to build networks prior to what we said in line 25, especially in the CNN setting. We will adjust the sentence in question to include references to earlier works and remove the word “first” in it.
> -	We are glad that the reviewer liked the use of the coloured box in line 157. We will add a title in a revised version of the paper.
>
> Finally, to answer the questions posed:
>
> 1)	We noted in Remark 4.17 of the paper that the pointwise non-linearities need to be odd to preserve the antisymmetric structure in the tensor. For normalisation, we need to normalise “symmetrically” across the relevant axes to maintain the antisymmetry e.g if we have an $n \times n$ antisymmetric matrix then we must normalise $T_{i,j}$ and $T_{j,i}$ in a paired way such that the updated $T_{i,j}$ and $T_{j,i}$ satisfy $T_{i,j} = - T_{j,i}$. This includes the case where we can divide every element in an antisymmetric tensor by a scalar, such as the maximum absolute value. We also note that residual connections maintain both the permutation equivariance and the antisymmetric structure.
> 2)	In practice, we can reasonably expect to compute with moderate $n$ (50 to 200) and to keep $k,l$ small, typically no greater than $3$, since computations start becoming intractable with higher $k$ and $l$. Much of our goal with the introduction of IO patterns was to mitigate memory issues caused by storing $n^l \times n^k$ weight matrices (that appear as a common bottleneck in related works), making the greater of $n^k$ and $n^l$ the main bottleneck on memory instead.

---

### Official Review · Reviewer_RLNR · 2025-11-01

**Soundness:** 3
**Presentation:** 3
**Contribution:** 3
**Rating:** 6
**Confidence:** 2

**Summary:**

This paper characterizes all linear permutation-equivariant maps between antisymmetric power spaces $\Lambda^{k}(\mathbb{R}^{n}) \to \Lambda^{\ell}(\mathbb{R}^{n})$ by giving an explicit basis for $\mathrm{Hom}{S_n} (\Lambda^{k} (\mathbb{R}^{n}), \Lambda^{\ell}(\mathbb{R}^{n}))$.
It provides a memory-efficient implementation that applies these maps directly to antisymmetric tensors without forming large unrolled matrices.
Experiments on $S_{8}$-equivariant and $S_{4}$-equivariant tasks show consistent gains over MLPs and standard permutation-equivariant baselines.

**Strengths:**

- The construction of linear permutation-equivariant maps for antisymmetric tensors is presented clearly and rigorously.
- The empirical results demonstrate consistent gains over MLP and permutation-equivariant baselines, achieving lower test MSE across data sizes while using fewer parameters.

**Weaknesses:**

The main weakness concerns experimental scope. Both the $S_4$-invariant and $S_8$-equivariant tasks are synthetic, so evidence for real-world applicability remains limited. It would be more convincing to evaluate the proposed antisymmetric permutation-equivariant maps on real datasets.

**Questions:**

Overall I think it is a good contribution to theoretically study the equivariant map for antisymmetric tensor. My main question is how the proposed maps translate to practice. Specifically, how could these maps be applied to relevant tasks? For example, in molecular/fermionic modeling or physics simulations?

---

> ### Author Response · Authors · 2025-11-17
>
> We thank the reviewer for their comments. We are glad that they “think it is a good contribution to theoretically study the equivariant maps for antisymmetric tensors”. We are pleased they recognised that our contribution is “presented clearly and rigourously”, and that our empirical results demonstrate our theoretical characterisation in practice.
>
> We welcome the reviewer’s feedback regarding the experimental scope. We agree that demonstrating the applicability of our model in a real-world setting strengthens the work. We have since evaluated our model on the real-world Wiki-RfA dataset, where we show that our model outperforms an unconstrained MLP in predicting local influence in subgraphs. Details of this experiment, together with a scaling and noise analysis that was requested by Reviewers DvEj and DPLT, are provided in our rebuttal to Reviewer DvEj.
>
> Finally, we believe that our real-world experiment demonstrates how our proposed maps translate to practice. In general, we consider a system with $n$ directed interacting elements and encode the interactions as an antisymmetric matrix/tensor, which serves as input. The maps that we have characterised propagate information to produce outputs at the node, edge, or scalar level. For example, they could potentially be used to predict a node’s influence in a social network or pairwise energies in a molecular structure, all while respecting the antisymmetric and permutation equivariant structure.

---

### Official Review · Reviewer_FmEQ · 2025-11-08

**Soundness:** 3
**Presentation:** 2
**Contribution:** 2
**Rating:** 2
**Confidence:** 4

**Summary:**

This paper presents a framework to construct permutation equivariant networks for antisymmetric tensors. By considering a canonical form where indices appear in lexicographical order, the authors are able to enumerate the basis of the transformation space, hence characterizing the equivariant transformations. Proof-of-concept experiments are presented in synthetic antisymmetric datasets to validate the constructions presented.

**Strengths:**

The authors go to great lengths to provide illustrations and explain their algorithms and the generating procedures. The constructions are intuitive and the overall structure of the paper is clean and logical.

**Weaknesses:**

The paper is definitely a difficult read and requires very careful and thorough pace. I found a lot of definitions taken for granted, which I think hurts exposition, especially for a general ML audience.

More on the general audience, I think it would have been extremely helpful to visualize (or give specific examples) of the vector spaces that are being considered. Concretely, (14) is a 3x3 matrix. How this matrix acts on elements of $\Lambda^2(\mathcal{R}^3)$ (to yield an element of $\Lambda^1(\mathcal{R}^3)$) is not trivial and concrete explanations about the practical implementation of the method would help presentation.

Finally, a lot of the motivation is coming from examples in the sciences, however the experiments don't have any experiments on real data, and the synthetic experiments are on very small models.

**Questions:**

1. How are zero length tuples defined? They are used in Proposition 3.4.
2. The proof of Proposition 4.1 is very difficult to read. As an example, $\\{E_{I, J}\\}_{I\in \Lambda[n]^l, J\in\Lambda[n]^k}$ is defined as having 1 in the $(I, J)$ position. However, these are tuples, so how is a position for a tuple of tuples defined? Standard ideas from linear vector spaces don't seem to suffice, and the background is missing.
3. More of a comment than a question, but the notation of $(k, l)$-antispherical bipartition obfuscates $n$, which seems problematic. It is not possible to deduce the vector space.
4. The algorithm above Proposition 4.9 is hard to parse, it's unclear why it is needed or what the intuition behind it is, and is combinatorial in order. Specifically for step 3, it is unclear what the order of a spider is.
5. There is no proof for Proposition 4.9, which seems like a central result of the paper.
6. In Appendix C an algorithm is presented, followed by the statement "Clearly, this generates all possible partitions". That is not obvious, and that statement is used in the sequel to count the number of bipartitions. A proof using induction (or any other proof tool) is necessary for statements like this.
7. The main text claims that the nonlinearity needs to be odd in order to preserve the antisymmetric nature, however ReLU is used for the experiments, which is clearly not odd. Moreover on the experiments, how is the generalization evaluated on differently sized tensors? Is $n$ the only thing that is changing, while the length of the tuples are kept the same? If so, how are the different indices casted to matrices?
8. Why is there no test accuracy reported for the first experiment, and why are the models trained for only 20 epochs, with such small model sizes?
9. Why are the running times not reported for the second experiment? In general, a complexity analysis for the method seems to be missing.

---

> ### Author Response · Authors · 2025-11-17
>
> We thank the reviewer for their comments and welcome the opportunity to clarify some points that may have caused confusion. We are pleased that the reviewer found our constructions to be “intuitive” and the overall structure of the paper to be “clean and logical”.
>
> We note that while the reviewer found the paper difficult to read, other reviewers found the presentation to be clear and engaging: Reviewer RLNR described it as being “presented clearly and rigourously”; Reviewer DPLT felt that the paper was “well-written” with the notation “clearly laid out so that no guessing is required”; and Reviewer DvEj described the key sections as being “particularly enjoyable”. However, we welcome the opportunity to improve the paper’s accessibility for a broader ML audience.
>
> In acknowledging that the topic is somewhat mathematically sophisticated, we are of the opinion that all definitions, constructions and proofs are provided in full detail – a view which the other reviewers appear to share. This ensures that the framework for the networks is rigourous and accessible to readers who follow the formal development of our results, even if some constructions are less familiar to a general ML audience. Beyond the reviewer’s questions, which we address below, if there are specific definitions or steps that the reviewer finds incomplete or taken for granted, then we would be happy to clarify them during the discussion phase.
>
> On the reviewer’s request for specific examples of the vector spaces under consideration and the practical implementation of our method, we note that we have provided a running example of the construction in Examples 4.2, 4.3, 4.7, and then another in Examples 4.11, 4.15 and 4.16 of the main paper. We have also provided additional examples of the calculations for both rolled and unrolled matrices in Appendix D. To further improve readability, we will add in an example of an element in $\Lambda^k(\mathbb{R}^{n})$ for some $k, n$ around line 100.
>
> In particular, regarding equation (14), applying this matrix to an element of $\Lambda^2(\mathbb{R}^{3})$ corresponds to standard matrix multiplication: a general element of $\Lambda^2(\mathbb{R}^{3})$, $x_{1,2}e_{1,2} + x_{1,3}e_{1,3} + x_{2,3}e_{2,3}$ (where $x_{i_1,i_2} \in \mathbb{R}$) becomes the column vector $[x_{1,2}, x_{1,3}, x_{2,3}]^T$ in the standard basis. We will add in a sentence that clarifies this for the general reader.
>
> To address concerns about the real-world applicability of our method, and to demonstrate our model’s scalability under noise, we have conducted two additional experiments, the details of which can be found in our rebuttal to Reviewer DvEj. They show further empirical support for our theoretical results.

---

> > ### Author Response · Authors · 2025-11-17
> >
> > To answer the questions posed:
> >
> > 1.	It is sufficient to note that $\Lambda^0(\mathbb{R}^{n}) = \mathbb{R}$. We will include this in line 99.
> >
> > 2.	Although the proof of Proposition 4.1 may appear challenging at first glance, we directly apply standard linear algebra concepts, and all necessary definitions are provided in the paper. In particular, $E_{I,J}$ is explicitly defined as a matrix with a single 1 and 0s elsewhere, with rows and columns indexed by the basis elements, which are tuples. This is analogous to the matrix representation of a map from $\mathbb{R}^n$ to $\mathbb{R}^n$ using the standard basis, where the rows and columns are indexed by elements of $\{1, \dots ,n\}$. More generally, the proof follows standard arguments for determining conditions for equivariant maps (using the definition of permutation equivariance given in Definition 3.3). We believe that any difficulty may arise from the density of symbols used rather than from any missing background. We will add a clarifying sentence on $E_{I,J}$ to make its definition fully explicit for readers who may be less familiar with these constructions.
> >
> > 3.	Regarding the definition of (k,l)-antispherical bipartitions in Definition 4.4, the use of $n$ is not needed. Each (k,l)-antispherical bipartition consists of a number of pairs, which we call blocks, so in our definition we had some $t$ blocks. $k$ and $l$ are fixed parameters, whereas $t$ can vary. In terms of calculating the $S_n$ equivariant linear matrices from $\Lambda^k(\mathbb{R}^{n})$ to $\Lambda^l(\mathbb{R}^{n})$, the proof shows that we only need to consider those (k,l)-antispherical bipartitions having at most n blocks (plus some extra conditions). The reviewer wrote “it is not possible to deduce the vector space.” We would appreciate clarification regarding which vector space is being referred to so that we can make sure that our explanation is fully clear.
> >
> > 4.	On the algorithm above Proposition 4.9: it is needed to establish the correspondence between (k,l)-antispherical bipartitions and orbits. Prior to this, we showed that each orbit corresponds to some (k,l)-antispherical bipartition having at most n blocks. Hence we needed to show that each (k,l)-antispherical bipartition having at most n blocks corresponds to an orbit, which is this algorithm. Consequently, it results automatically in Proposition 4.9 and its proof. The intuition behind it is that the elements of the orbit can be read off the diagram by labelling the blocks with numbers and pushing the labels to the edges, resulting in a tuple $I$ from the top of the diagram and $J$ from the bottom. However, we need to reorder the spiders to make the block labels be in increasing numerical order so that the resulting tuples $I$ and $J$ are in increasing numerical order (i.e so that they are elements of $\Lambda[n]^l$ and $\Lambda[n]^k$, respectively). To improve clarity, we will edit step 3 to say “If necessary, reorder the spiders so that the labels of their central red nodes are in increasing numerical order from left to right.”
> >
> > 5.	We addressed this point in our response to Question 4.
> >
> > 6.	We respectfully disagree that the proof of Lemma C.1 is not obvious from the construction, or that induction is needed to prove the statement. We note that the proof follows directly from the construction of the tableaux and the way we colour them in (an example of this colouring in is provided just above in (31)). The proof of this lemma is provided below its statement in lines 844-849.
> > 7.	We can use ReLU in these experiments because $\Lambda^1(\mathbb{R}^{n}) = \mathbb{R}^{n}$, i.e the basis is indexed by a tuple of length one. Regarding generalisation: for the fixed values of $k, l$ ($k = 2, l = 1$), we generate 2000 test cases for each value of $n$ to obtain the accuracy. On “casting the different indicies to matrices”, we note that we do not use matrix implementations to construct the networks but use IO patterns instead, which provide the flexibility of using different values of $n$ for fixed $k$ and $l$. This experiment demonstrates in practice the theoretical property that training on one value of $n$ generalises to larger $n$ by adjusting $n$ in the IO pattern (see Remark 5.3).
> > 8.	We report Test MSE in Table 1 for the first experiment (see the number provided for $n =  8$). Test accuracy is not meaningful here as it is a regression problem. We trained for 20 epochs as this was sufficient for our model to converge. When we extended training to, say, 100 epochs, it did not improve the performance of the other models as they still failed to learn. We therefore limited the plots to 20 epochs to avoid long, uninformative tails. We intentionally used small model sizes in the first experiment as our goal was to provide a proof of concept for how our proposed layer behaves. This setup allowed us to clearly observe the layer’s effect without the added complexity of larger architectures.

---

> > > ### Author Response · Authors · 2025-11-17
> > >
> > > 9.	We didn’t include running times as our focus was on how performance scales with data across different data sizes and runs, not on how long it takes. We also note that the IO pattern implementation reduces memory usage compared with naive matrix multiplication, which indirectly improves efficiency as $n$, $k$, and $l$ grow.
> > >
> > > In answering questions 7-9, we wish to highlight that the new experiments we provided in our rebuttal to Reviewer DvEj largely supersede these concerns, offering direct evidence of our model’s practical applicability and its performance across different settings.

---

### Meta-Review · Area_Chair_GEZG · 2026-01-05

**Summary:**

All reviewers agree that the paper’s main strength lies in its theoretical development: it gives a full characterization of linear permutation-equivariant maps between antisymmetric tensor spaces and proposes a memory-efficient implementation. However, they diverge on how well this fits ICLR’s balance of theory, clarity, and empirical validation.

Reviewer FmEQ (2) finds the paper very difficult to read for a general ML audience, with important definitions and constructions taken for granted, several proofs and algorithms hard to parse, and key steps (e.g., Proposition 4.1, the algorithm above Proposition 4.9, and Lemma C.1) insufficiently justified from their perspective. They are also concerned about the lack of real data, the very small synthetic setups, missing complexity/runtime analysis, and some inconsistencies (e.g., odd nonlinearity vs. ReLU use).

Reviewer RLNR (6) and DPLT (8) view the theoretical contribution as strong, elegant, and clearly presented, but both flag experimental scope and relevance as a main weakness: experiments are initially only synthetic, small-scale, and do not convincingly demonstrate real-world impact or scalability. DPLT also questions the novelty relative to existing mathematics on exterior powers and Hom spaces and notes that the application domain is niche for the broader ML community.

Reviewer DvEj (4→6) explicitly praises the theory but initially considers the empirical part too weak for ICLR: very small models, toy datasets, and limited evidence about robustness and scalability. They emphasize that, for a machine learning conference, some stronger empirical demonstration is needed to complement the strong theory.

Taken together, the concerns that drive my decision are: (i) serious accessibility and clarity issues for at least one reviewer; (ii) uncertain novelty and positioning with respect to existing representation-theoretic results; and (iii) limited empirical validation and unclear practical impact, even after the new experiments.

**Reviewer Concerns:**

The authors’ rebuttal successfully addresses several important points:

- In response to RLNR and DvEj, the authors add a real-world experiment on the Wiki-RfA social network dataset and a scaling and noise analysis on larger synthetic graphs, showing that the antisymmetric model can outperform an unconstrained MLP and behaves reasonably as graph size and noise increase.

- They clarify multiple technical details: how matrices act on antisymmetric tensors in practice, how non-linearities and normalisation interact with antisymmetry, realistic ranges of $n, k, \ell$, and why certain implementation choices (e.g., IO patterns) reduce memory usage.

- For DPLT, they sharpen the novelty claim, arguing that while exterior powers and Hom spaces are classical, the characterization of $\mathrm{Hom}_{S_n}$​​ (permutation-equivariant maps) and the antispherical bipartitions/IO pattern construction are new in this specific equivariant setting.

These additions and clarifications clearly strengthen the submission and were sufficient for DvEj to raise their score and for RLNR to remain positive. However, several concerns remain only partially resolved:

- From FmEQ’s perspective, the paper remains mathematically dense and difficult for a typical ICLR audience. While the authors argue that all definitions are present, the core issue is accessibility, not just completeness: key proofs, combinatorial algorithms, and diagrammatic constructions are still hard to follow without substantial background, and some steps continue to look “obvious” only to a specialized reader.

- Regarding novelty and positioning, while the authors explain that HomSn_{S_n}Sn​​ spaces and antispherical bipartitions are new in this context, the paper still does not fully bridge the gap between deep representation theory and ML readership: it remains somewhat unclear how much is fundamentally new mathematics versus a careful repackaging and implementation of known representation-theoretic ideas for antisymmetric tensors.

- On the empirical side, the new experiments help, but they remain relatively small-scale: the real-world task is low-dimensional, and the synthetic graphs, while larger, still do not demonstrate the method’s behavior in the kinds of large, high-dimensional regimes typical of modern ML applications. The paper still feels primarily like a theory paper with proof-of-concept experiments, and its practical impact for the broader ICLR community remains uncertain.

Because of these outstanding issues, particularly accessibility and limited demonstrated impact, I do not think the rebuttal is sufficient to bring the paper clearly above the ICLR bar.

**Reviewer Scores:**

**Reviewer FmEQ (2):** FmEQ’s main issues concern readability, missing or opaque explanations, and lack of substantial empirical validation. The rebuttal addresses their specific questions point by point and adds new experiments, so I could imagine a slight softening of their view (e.g., from a very strong reject toward a more moderate reject). However, the fundamental concern, that the paper is too hard to digest for a general ML audience and still light on convincing, large-scale empirical evidence, would likely remain. I therefore expect FmEQ to stay in the reject range, perhaps with slightly higher confidence in the correctness but not in suitability for ICLR.

**Reviewer RLNR (6):** RLNR was already marginally positive, with the main concern being the absence of real-world experiments. The rebuttal directly addresses this by adding the Wiki-RfA experiment, and RLNR’s comments indicate they see the work as a good theoretical contribution. I expect RLNR would maintain their 6 (marginally above threshold) after discussion, perhaps with slightly higher confidence, but still not pushing strongly for acceptance.

**Reviewer DPLT (8):** DPLT is very positive about the mathematics and writing style, and their main concerns--novelty clarification, additional analysis, and motivation by applications--are partially addressed in the rebuttal and new experiments. I expect DPLT would keep an 8 and remain a strong proponent of acceptance, emphasizing the elegance and rigor of the contribution.

**Reviewer DvEj (4 → 6):** DvEj explicitly states in their follow-up comment that the new experiments “sufficiently address” their earlier empirical concerns and raises their score to 6, with a positive stance. After full discussion, I expect them to remain at 6, supporting acceptance but not as strongly as DPLT.

In summary, after rebuttal and likely discussion, the score profile would be something like: one strong reject, one clear accept, and two weak accepts. Given the outstanding issues around accessibility, limited empirical breadth, and uncertain practical impact, and weighing the strong negative view from a confident reviewer against the relatively borderline nature of the positive scores, I side with the more cautious assessment and recommend rejection.

---

### Decision · Program_Chairs · 2026-01-26

Reject